# Debate or Vote: Which Yields Better Decisions in Multi-Agent Large Language Models?

**Hyeong Kyu Choi**    **Xiaojin Zhu**    **Sharon Li**[*]
Department of Computer Sciences, University of Wisconsin-Madison
{froilanchoi, jerryzhu, sharonli}@cs.wisc.edu

## Abstract

Multi-Agent Debate (MAD) has emerged as a promising paradigm for improving the performance of large language models through collaborative reasoning. Despite recent advances, the key factors driving MAD's effectiveness remain unclear. In this work, we disentangle MAD into two key components–Majority Voting and inter-agent Debate–and assess their respective contributions. Through extensive experiments across seven NLP benchmarks, we find that Majority Voting alone accounts for most of the performance gains typically attributed to MAD. To explain this, we propose a theoretical framework that models debate as a stochastic process. We prove that it induces a martingale over agents' belief trajectories, implying that debate alone does not improve expected correctness. Guided by these insights, we demonstrate that targeted interventions, by biasing the belief update toward correction, can meaningfully enhance debate effectiveness. Overall, our findings suggest that while MAD has potential, simple ensembling methods remain strong and more reliable alternatives in many practical settings. Code is released in https://github.com/deeplearning-wisc/debate-or-vote.

## 1    Introduction

*"Out of intense complexities, intense simplicities emerge."*

— W. CHURCHILL

Throughout history, humans have relied on deliberation to resolve ambiguity, challenge assumptions, and seek better answers. From courtrooms and panels to scientific collaborations, group reasoning plays a central role in decision-making. This process—where individuals reflect, revise, and converge through interaction—has long been seen as a hallmark of intelligent behavior. Inspired by this, recent work has explored whether large language models (LLMs) might similarly benefit from structured interaction. Multi-Agent Debate (MAD) has emerged as a popular framework: multiple LLM agents are prompted to discuss a shared question, each updating their answer based on the responses of their peers [1–6]. The hope is that, like human deliberation, such interaction will improve reasoning and lead to better outcomes.

At its core, MAD integrates two key ingredients: the use of multiple agents ("Multi-Agent") and their interaction through iterative discussions ("Debate"). Recent work has introduced increasingly sophisticated variants—ranging from diverse communication protocols [3, 7, 8], designing efficient and effective system architectures [1, 2, 9, 10], and assigning varied roles or personas to agents [11–13]. Despite these advances, the underlying mechanisms behind MAD's effectiveness remain unclear. A natural step toward understanding MAD's performance is to disentangle the contribution of each component—***are the gains primarily due to meaningful communication between agents, or simply the result of aggregating multiple outputs?*** Answering this question is important because it

---

[*]Corresponding author

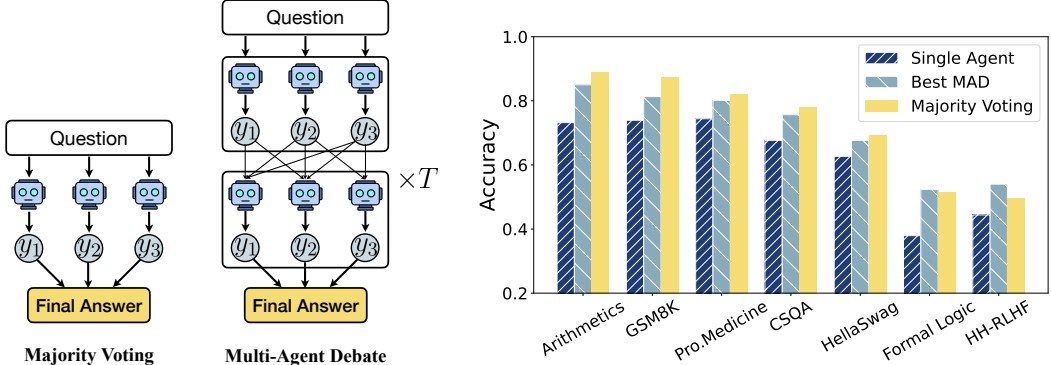

Figure 1: Majority Voting vs. MAD overview.

Figure 2: Majority Voting is the main contributor to MAD.

informs whether the growing complexity of MAD design is justified by tangible benefits. If most of the performance gain stems from ensembling—*i.e.*, aggregating diverse outputs from independent agents—then simpler methods like majority voting may suffice, avoiding additional computational and architectural overhead (see Figure 1 for visual comparison).

To better understand the relative contributions of ensembling *vs.* interaction, we conduct an extensive empirical study quantifying each component's effect. Specifically, we measure the contribution of the "Multi-Agent" component using the performance achieved through Majority Voting, *i.e.*, the aggregated output of agents before any debate rounds occur. We then compare this baseline to the final performance after multiple rounds of "Debate", allowing us to isolate the additional benefit introduced by inter-agent communications. Surprisingly, we find that Majority Voting accounts for most of the performance gains in MAD. In fact, in most cases, *majority voting **without any debate** performs on par with MAD*, as seen in Figure 2. To ensure the broad applicability of our findings, our evaluation spans seven diverse benchmarks across multiple tasks and models.

Beyond empirical observations, we introduce a theoretical framework in Section 4 that rigorously explains how agents' uncertainty and belief updates shape collective decision-making in both voting and debate. At its core, the framework models each agent as a stochastic process governed by a Dirichlet-Compound-Multinomial (DCM) distribution, capturing internal uncertainty through a Dirichlet belief prior and output randomness via Multinomial sampling. This closely mirrors the behavior of real-world LLMs, which produce different outputs for the same question due to uncertainty and stochastic generation process (*e.g.*, via temperature or nucleus sampling). Within this framework, we characterize MAD as a Bayesian posterior belief update process and prove that it induces a ***martingale*** over agents' belief in the correct answer—meaning the expected belief remains unchanged over debate rounds. This implies that debate itself does not systematically improve or degrade beliefs on average; rather, belief evolution is driven by stochastic peer influence. In other words, we prove formally that *majority vote does essentially all the work*, which explains our empirical findings.

Our theoretical framework further sheds light on the new design principles to improve MAD (Section 5). In particular, it highlights the importance of controlling the martingale by biasing belief updates toward correct signals during debate. We operationalize this insight through several interventions, where correct responses exert more influence than misleading ones, leading to improvements over standard MAD. We summarize our contributions and significance as follows:

1. We comprehensively demonstrate that Majority Voting is as effective as Multi-Agent Debate, when evaluated across seven representative benchmark datasets. We further expand our investigation to more general MAD settings, including configurations with larger and more capable agents, heterogeneous agent populations, and open-ended natural language tasks.

2. We develop a *new theoretical framework* that reveals majority voting's success probability, and rigorously characterizes multi-agent debate as a martingale process. This framework lays a principled foundation for future work to better understand MAD systems.

3. Our theoretical analysis informs that debate alone does not improve beyond majority voting. By designing strategies that help preserve correct responses across debate rounds, we achieve

notable improvements in multi-agent debate performance. This sheds light on future research to effectively improve MAD systems.

## 2 Preliminaries

**Multi-Agent Debate**   is a collaborative framework in which multiple language model agents engage in structured interaction—typically in the form of iterative exchanges or discussions—to solve a task such as question answering or text generation [1–6]. In a typical MAD protocol, each agent independently generates an initial response and then engages in a series of debate rounds. At round $t$, an agent receives the original question along with responses from its peers at round $t − 1$, prompting the model to update its response accordingly. This iterative process is designed to leverage diverse reasoning paths and peer wisdom, potentially enhancing the overall decision quality. After all rounds of debate, the final answer is typically derived through an aggregation mechanism, such as majority voting. Specific prompts are provided in Appendix B.1.

**Debate *vs.* Voting: Formalization.**   Let $\mathcal{X}$ denote the input space (e.g., natural language questions), and $\mathcal{Y}$ the output space (e.g., free-form or multiple-choice answers). We consider a set of $N$ language model agents, denoted by $\{a_1, \ldots, a_N\}$, where each agent defines a stochastic function $f_i : \mathcal{X} \to \mathcal{Y}$ that produces an initial response $y_{i,0} \sim f_i(x)$ for input $x \in \mathcal{X}$.

In the *Majority Voting* setting, the initial responses $\{y_{i,0}\}_{i=1}^N$ are directly aggregated using a voting function $\mathcal{V} : \mathcal{Y}^N \to \mathcal{Y}$ to obtain the final prediction, typically returning the most frequent answer:

$$y_0 = \mathcal{V}(y_{1,0}, \ldots, y_{N,0}).$$

In contrast, *Multi-Agent Debate* introduces $T$ rounds of iterative communication. We formalize the communication structure of debate as an undirected graph $\mathcal{G}$, where each node corresponds to an agent and edges indicate which agents observe one another. At round $t \geq 1$, each agent $a_i$ observes responses from a set of neighboring agents at the previous round and updates its answer accordingly. We define the response set of neighbors available to agent $a_i$ at round $t$ as:

$$\mathcal{R}_i^{(t)} = \{y_{j,t-1} \mid j \in \mathcal{N}(i)\},$$

where $\mathcal{N}(i) \subseteq \{1, \ldots, N\}$ is the index set of neighbors observable to agent $a_i$, including itself (*e.g.*, in the fully connected setting, $\mathcal{N}(i) = \{1, \ldots, N\}$). The response update is given by:

$$y_{i,t} = \mathcal{D}\left(x; \mathcal{R}_i^{(t)}\right),$$

where $\mathcal{D}$ denotes a single round of debate. The iterative debate process over $T$ rounds can be expressed as a function composition:

$$y_{i,T} = (\mathcal{D} \circ \mathcal{D} \circ \cdots \circ \mathcal{D})(x; \mathcal{R}_i) = \mathcal{D}^{(T)}(x; \mathcal{R}_i).$$

The final aggregated output after $T$ rounds is:

$$y_T = \mathcal{V}(y_{1,T}, y_{2,T}, \cdots, y_{N,T}).$$

We adopt the simultaneous-talk protocol [3], where all agents update in parallel based on the previous round's responses. Following the common setup in prior works, we focus on homogeneous agent settings, *i.e.*, all agents share the same underlying model architecture or behavior. This allows us to isolate the effect of inter-agent communication and contrast MAD directly with simple majority voting. Our goal is to contrast the performance of MAD against simple majority voting and assess whether iterative inter-agent communication provides measurable improvements beyond ensembling alone. We will extend to the heterogeneous setting in Section 6.

## 3 Is Debate Really Necessary? A Closer Look at Debate *vs.* Voting

Multi-agent debate is often regarded as a promising mechanism for enhancing LLM performance via collaborative deliberation. *But how much of its effectiveness truly comes from the debate itself—and how much is simply due to aggregating multiple answers*? To address this question, we dissect MAD into two components—multi-agent ensembling and inter-agent communication—and present empirical evidence revealing that simple majority voting accounts for most of the observed gains. We begin by explaining the experimental setup in the following section.

Table 1: **Majority Voting vs. Multi-Agent Debate.** Benchmark performances are measured in Accuracy.

| Methods | Qwen2.5-7B-Instruct | | | | | | | |
| --- | --- | --- | --- | --- | --- | --- | --- | --- |
| | Arithmetics | GSM8K | MMLU (Pro.Med.) | MMLU (Form.Log.) | HellaSwag | CommonSense QA | HH-RLHF | Average |
| **Single-Agent** | | | | | | | | |
| Single-agent baseline | $0.8140 \pm .04$ | $0.8713 \pm .00$ | $0.7868 \pm .01$ | $0.4905 \pm .03$ | $0.7880 \pm .01$ | $0.8153 \pm .01$ | $0.4773 \pm .01$ | 0.7205 |
| **Multi-Agent** | | | | | | | | |
| Decentralized MAD ($T=2$) | 0.7600 | 0.8867 | 0.8051 | **0.5556** | 0.8033 | **0.8567** | 0.4967 | 0.7377 |
| Decentralized MAD ($T=3$) | 0.6700 | 0.8533 | 0.8051 | 0.5000 | 0.8000 | 0.8500 | 0.5000 | 0.7112 |
| Decentralized MAD ($T=5$) | 0.6700 | 0.8333 | 0.8051 | 0.4762 | 0.8000 | 0.8433 | **0.5067** | 0.7050 |
| Sparse MAD ($T=2$) | 0.8400 | 0.9033 | 0.8051 | 0.4762 | 0.7967 | 0.8367 | 0.4733 | 0.7330 |
| Sparse MAD ($T=3$) | 0.8100 | 0.8833 | **0.8162** | 0.4365 | 0.7967 | 0.8367 | 0.4733 | 0.7218 |
| Sparse MAD ($T=5$) | 0.7900 | 0.8700 | 0.8088 | 0.4365 | 0.7900 | 0.8333 | 0.4833 | 0.7160 |
| Centralized MAD ($T=2$) | 0.4300 | 0.7300 | **0.8162** | 0.4762 | 0.8100 | **0.8567** | 0.4667 | 0.6551 |
| Centralized MAD ($T=3$) | 0.4800 | 0.7367 | **0.8162** | 0.4603 | 0.8100 | 0.8500 | 0.4733 | 0.6609 |
| Centralized MAD ($T=5$) | 0.5500 | 0.7200 | 0.8125 | 0.4444 | **0.8133** | 0.8467 | 0.4833 | 0.6672 |
| **Majority Voting** | **0.9900** | **0.9400** | 0.7941 | 0.5397 | 0.8033 | 0.8300 | 0.4867 | **0.7691** |

| Methods | Llama3.1-8B-Instruct | | | | | | | |
| --- | --- | --- | --- | --- | --- | --- | --- | --- |
| | Arithmetics | GSM8K | MMLU (Pro.Med.) | MMLU (Form.Log.) | HellaSwag | CommonSense QA | HH-RLHF | Average |
| **Single-Agent** | | | | | | | | |
| Single-agent baseline | $0.7320 \pm .03$ | $0.7393 \pm .01$ | $0.7441 \pm .01$ | $0.3794 \pm .02$ | $0.6267 \pm .03$ | $0.6767 \pm .01$ | $0.4440 \pm .02$ | 0.6203 |
| **Multi-Agent** | | | | | | | | |
| Decentralized MAD ($T=2$) | 0.8200 | 0.7933 | 0.7868 | **0.5238** | 0.6767 | 0.7267 | 0.5233 | 0.6929 |
| Decentralized MAD ($T=3$) | 0.8300 | 0.7933 | 0.7684 | 0.5000 | 0.6300 | 0.7033 | 0.5267 | 0.6788 |
| Decentralized MAD ($T=5$) | 0.8500 | 0.7800 | 0.7463 | 0.5000 | 0.6300 | 0.7000 | 0.5267 | 0.6761 |
| Sparse MAD ($T=2$) | 0.8500 | 0.8133 | 0.8015 | 0.4683 | 0.6767 | 0.7567 | 0.5267 | 0.6990 |
| Sparse MAD ($T=3$) | 0.8500 | 0.7967 | 0.7831 | 0.4206 | 0.6233 | 0.7467 | 0.5333 | 0.6791 |
| Sparse MAD ($T=5$) | 0.8500 | 0.7667 | 0.7868 | 0.4365 | 0.6233 | 0.7233 | **0.5400** | 0.6752 |
| Centralized MAD ($T=2$) | 0.7300 | 0.6400 | 0.6949 | 0.3810 | 0.6000 | 0.7400 | 0.4800 | 0.6094 |
| Centralized MAD ($T=3$) | 0.7700 | 0.6200 | 0.6507 | 0.3730 | 0.6133 | 0.7200 | 0.4900 | 0.6053 |
| Centralized MAD ($T=5$) | 0.7300 | 0.5933 | 0.5846 | 0.3413 | 0.5800 | 0.6967 | 0.4433 | 0.5670 |
| **Majority Voting** | **0.8900** | **0.8733** | **0.8199** | 0.5159 | **0.6933** | **0.7800** | 0.4967 | **0.7242** |

## 3.1 Experimental Setup

**Baselines.** The key distinction among multi-agent debate methods typically lies in the design of the debate function $\mathcal{D}$, particularly in how agents communicate and the roles they assume. To comprehensively evaluate these variations, we consider the following representative approaches: (1) *Decentralized MAD* [2], where each agent observes all other agents' responses from the previous round. (2) *Sparse MAD* [10], a variant of Decentralized MAD with a sparse communication topology to enhance efficiency. (3) *Centralized MAD* [14], where a central agent aggregates peer responses and generates the updated response at each round. (4) *Majority Voting*, which selects the final answer by aggregating initial responses from multiple agents **without any debate**. This can be viewed as a special case with $T = 0$. For all the multi-agent approaches, we adopt $N = 5$ in our main comparison and will ablate on the effect of $N$. For single-agent baselines, we average across 5 independent runs.

**Benchmarks.** Following previous MAD literature, we focus on solving six natural language question answering tasks by conducting extensive evaluations across benchmark datasets: (1) *Arithmetics*, (2) *Mathematical Reasoning* (Grade School Math 8k [15]), (3) *Factual Question Answering* (MMLU Professional Medicine and Formal Logics [16, 17]), (4) *Natural Langauge Inference* (HellaSwag [18]), (5) *Commonsense Reasoning* (CommonsenseQA [19]), and (6) *Alignment Labeling* (HH-RLHF [20]), where we adopt the "AI labeler alignment" practice [21], similar to [10]. For fairness in comparison, all baselines are evaluated on the same data subsets. More details are provided in Appendix A.2.

## 3.2 Key Observations

**Majority voting is surprisingly strong.** In Table 1, we compare the performance of single-agent, MAD, and majority voting approaches across seven benchmark datasets using the Qwen2.5-7B-Instruct [22] and Llama3.1-8B-Instruct [23] models. Consistent with the typical choice of existing literature, we compare 2- and 3-round, along with a prolonged 5-round debate setting among five agents. Interestingly, while MAD consistently outperforms the single-agent baseline, it does not

reliably surpass the much simpler majority voting strategy. *Notably, in most cases, majority voting performs on par with MAD.* To further assess the impact of model capacity, we additionally evaluate the more capable Qwen2.5-32B-Instruct model in Section 6. Although overall performance improves in the MAD setting, the majority voting strategy continues to account for most of the performance gains. These findings suggest that the effectiveness of MAD is largely driven by model ensembling, rather than the iterative debate process itself.

To gain deeper insight into the effect of MAD components, we present an ablation study in Figure 3. It illustrates the effect of varying the number of Qwen2.5 agents participating in each round of debate, from $N = 1$ to $N = 5$. Overall, increasing the number of agents generally leads to improved performance. The trend suggests that MAD's effectiveness may stem primarily from the ensemble effect of multiple agents. Our next section formalizes our observations.

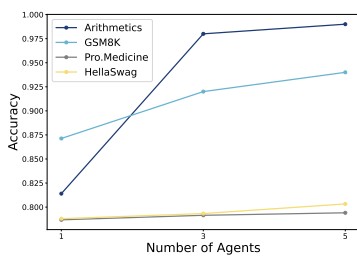

Figure 3: Accuracy improves with more agents.

# 4 Theoretical Analysis

To better understand the dynamics underlying our empirical findings in Section 3, we now turn to a formal analysis of multi-agent debate and majority voting. Our theoretical framework allows us to capture how agent uncertainty and belief updates shape the collective decision-making in both debate and voting, grounded in Bayesian principles. Specifically, for a given input question, we consider a population of $N$ agents, each generating a response from a finite set $\mathcal{A}$, which may represent multiple-choice question options or a set of plausible completions for open-ended tasks. We model each agent as an idealized generative model governed by a Dirichlet-Compound-Multinomial (DCM) distribution. *This closely mirrors how practical LLM systems produce different outputs for the same question due to uncertainty and sampling variability.* In particular, the Dirichlet prior captures the agent's internal belief over possible answers, while the Multinomial models the stochastic generation process (*e.g.*, via temperature or nucleus sampling). This distribution is thus a natural choice because it encapsulates both internal uncertainty and output randomness, while also providing a principled Bayesian framework for belief updates across debate rounds—enabling analytical study of dynamics during the debate process. Below we provide the mathematical details of the DCM model.

**Definition 1. (Agent Response Generation via DCM)** *At round $t$, each agent $i$ is associated with a belief vector $\boldsymbol{\alpha}_{i,t} = (\alpha_{i,t}^{(1)}, \ldots, \alpha_{i,t}^{(K)}) \in \mathbb{R}_+^K$, where each entry $\alpha_{i,t}^{(k)}$ reflects the agent's belief in response option $k \in \mathcal{A}$. To generate a response $y_{i,t}$, the agent follows a two-step process:*

$$\textit{(Belief sampling)} \quad \boldsymbol{\theta}_{i,t} \sim \mathrm{Dirichlet}(\boldsymbol{\alpha}_{i,t}),$$
$$\textit{(Response generation)} \quad y_{i,t} \sim \mathrm{Categorical}(\boldsymbol{\theta}_{i,t}).$$

*The marginal probability of generating any particular response $y_{i,t} \in \mathcal{A}$—after integrating out the randomness in $\boldsymbol{\theta}_{i,t}$—is given by $P(y_{i,t} = k \mid \boldsymbol{\alpha}_{i,t}) = \alpha_{i,t}^{(k)} / \sum_{j \in \mathcal{A}} \alpha_{i,t}^{(j)}$.*

Before analyzing the dynamics of debate, we first consider the base case where agents respond independently without interaction. The following characterizes the success probability of majority voting under this condition, based solely on the agents' homogeneous initial beliefs, $\boldsymbol{\alpha}_{i,0} = \boldsymbol{\alpha} = (\alpha^{(1)}, \ldots, \alpha^{(K)})$. Without loss of generality, let answer index 1 be the correct option. We assume that the correct answer has the largest belief $\alpha^{(1)}$, and all the other beliefs are ordered such that $\alpha^{(1)} > \alpha^{(2)} \geq \cdots \geq \alpha^{(K)}$.

**Theorem 1. (Majority Voting Success Probability)** *Let $\bar{\boldsymbol{\theta}} = (\bar{\theta}^{(1)}, \ldots, \bar{\theta}^{(K)}) = \boldsymbol{\alpha} / \sum_{j=1}^{K} \alpha_j$ denote the mean of the Dirichlet distribution, $\mathrm{Dirichlet}(\alpha)$, and define the margin $\Delta := \bar{\theta}_1 - \bar{\theta}_2$. If $N > K/\Delta^2$, then the probability that majority voting selects the answer 1 is lower bounded as:*

$$\mathbb{P}(y_{\mathrm{mv}} = 1) \geq 1 - \exp\left(-N\left(\frac{\Delta}{\sqrt{K}} - \frac{1}{\sqrt{N}}\right)^2\right).$$

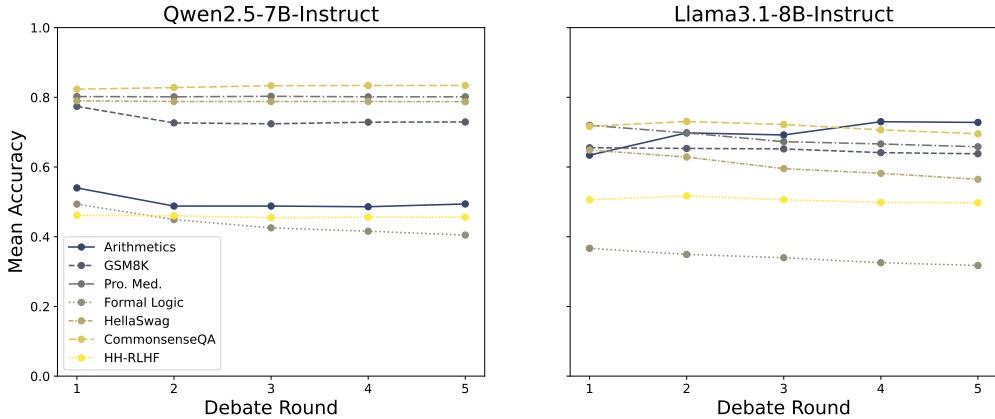

Figure 4: Martingale process of the mean agent accuracy across debate rounds.

**Remark 1.** This result highlights the *magnifying effect* of majority voting: even when the correct answer is only marginally more likely than the alternative answers, the lower bound on the success probability asymptotically approaches 1 as $N$ scales. Notably, this holds even when $\bar{\theta}_1 \ll \frac{1}{2}$, as long as it remains the most probable choice option. In practice, we recognize that MAD systems often operate with a small number of agents due to computational constraints. We provide a sharper analysis specialized to this realistic regime that applies to arbitrary $N$ in Theorem 1.A (Appendix C), without constraining its size. This complementary result provides insights into the reliability of majority voting in more practical, resource-constrained settings.

Next, we analyze the multi-agent debate performance by formalizing how each agent's belief $\boldsymbol{\alpha}_{i,t}$ evolves through debate. Specifically, each agent observes its neighbors' responses and performs a Bayesian posterior update of its belief accordingly.

**Definition 2. (Bayesian Belief Update from Neighbor Responses)** Let $\{y_{j,t-1} \mid j \in \mathcal{N}(i)\}$ be the set of responses observed by agent $i$ from its neighbors $\mathcal{N}(i)$ at round $t$. These responses induce a count vector $\mathbf{c}_{i,t} = (c_{i,t}^{(1)}, \ldots, c_{i,t}^{(K)}) \in \mathbb{N}^K$, where $c_{i,t}^{(k)}$ denotes the number of neighbors who selected response $k$. Then, the agent updates its Dirichlet parameter as: $\boldsymbol{\alpha}_{i,t} = \boldsymbol{\alpha}_{i,t-1} + \mathbf{c}_{i,t}$.

Under this formulation, each round of multi-agent debate corresponds to a Bayesian update step under the conjugacy of the Dirichlet-Multinomial model.

**Lemma 1. (Bayesian Conjugacy in Multi-Agent Debate)** At round $t$, after observing responses from its neighbors $\mathcal{N}(i)$, the agent $i$ aggregates these into a count vector $\mathbf{c}_{i,t}$ as in Definition 2. Then, by Bayesian conjugacy, the posterior distribution over $\boldsymbol{\theta}_{i,t}$ remains Dirichlet:

$$\boldsymbol{\theta}_{i,t} \mid \{y_{j,t-1}\}_{j \in \mathcal{N}(i)} \sim \mathrm{Dirichlet}(\boldsymbol{\alpha}_{i,t-1} + \mathbf{c}_{i,t}).$$

Then, for an agent $i$ with neighbors $\mathcal{N}(i)$, let $p_{i,t} := \bar{\theta}_{i,t}^{(1)}$ denote its belief in the correct answer at debate round $t$. Under Bayesian conjugacy,

$$p_{i,t} := \bar{\theta}_{i,t}^{(1)} = \frac{\alpha_{i,t}^{(1)}}{\sum_{j=1}^{K} \alpha_{i,t}^{(j)}} = \frac{\alpha_{i,t-1}^{(1)} + c_{i,t}^{(1)}}{\sum_{j=1}^{K}(\alpha_{i,t-1}^{(j)} + c_{i,t}^{(j)})}.$$

This agent belief's evolution throughout debate forms a martingale process (Theorem 2), with full proof in Appendix D.

**Theorem 2. (Martingale Behavior of Multi-Agent Debate)** For any agent $i$ at round $t > 0$, if

$$\frac{1}{|\mathcal{N}(i)|} \sum_{j \in \mathcal{N}(i)} p_{j,t-1} = p_{i,t-1}, \tag{1}$$

then sequence $\{p_{i,t}\}_{t \geq 0}$ forms a martingale. That is, the expected belief at the next round equals the current belief:

$$\mathbb{E}[p_{i,t} \mid \boldsymbol{\alpha}_{t-1}] = p_{i,t-1}.$$

**Theoretical insights: Majority vote does essentially all the work.**   This theorem highlights a fundamental property of multi-agent debate under Bayesian update of the DCM model: the agent's belief in the correct answer behaves like a *martingale*—that is, its expected value remains unchanged across rounds. This result is closely related to the classical Pólya Urn scheme [24]. Intuitively, this means that debate itself does not systematically improve or degrade an agent's belief on average; instead, belief updates are driven entirely by the stochastic influence of peer responses. While some debate trajectories may lead to stronger belief in the correct answer (*i.e.*, *correction*), others may lead to weaker belief (*i.e.*, *subversion*). While these local fluctuations affect the posterior counts, the expected belief in the correct answer remains equal to the initial $p_{i,0} = \bar{\theta}_{i,0}^{(1)}$, without any debate, when agents are homogeneous and fully-connected (Appendix D.3.1). This implies that, under our theoretical model, debate alone does not necessarily improve the initial accuracy–*majority voting accounts for the primary performance gains*. Our theory thus aligns well with our empirical findings in Section 3. Furthermore, in Appendix D.4, we extend our analysis to a more general setting in which each agent's expected belief evolves over the course of the debate, such that the condition in Theorem 2 may no longer hold. In this context, we introduce the notion of "collective intelligence" and demonstrate that it forms a martingale process even with heterogeneous agents.

**Martingale behavior is supported empirically.**   We empirically examine whether the sequence $\{p_{i,t}\}_{t \geq 0}$ exhibits martingale behavior. For each benchmark and debate round $t$, we estimate $p_t$ as the mean accuracy of the five agents[2]. As shown in Figure 4, *the resulting trajectories are essentially flat, which is consistent with the theoretical property that the expected value of a martingale remains unchanged over time*. Raw values for mean accuracy are provided in Table 6 (Appendix E).

**Generalized interpretation of the Bayesian update step.**   In Lemma 1 and Theorem 2, we define the update dynamics in terms of the count vector $\mathbf{c}_{i,t}$. Our framework can be generalized to open-ended tasks and capture the heterogeneous influence of each agent's response, with minor interpretational adjustments. For open-ended tasks, although responses are not strictly countable in categorical form, they can be represented in distributional or similarity-based spaces. Here, the count vector can be viewed more broadly, *e.g.*, as a soft histogram over clustered response types, an embedding-based semantic agreement measure, or a weighted similarity score between textual outputs. In such settings, "consensus" is better defined semantically rather than symbolically: if agents independently produce explanations or rationales that are semantically aligned, this can be considered consensus even when surface forms differ. This notion can be operationalized using thresholds on embedding similarity, Levenshtein distance, or overlap in reasoning chains, among other metrics.

## 5   How Does Theory Inform Improved Design of MAD?

The martingale property in our theory reflects a neutral expectation over time, underscoring that without additional bias towards correct signals, debate alone does not guarantee convergence to the truth. Any additional benefit arises from local asymmetries in the observed stochastic process $\{p_t\}_{t \geq 0}$. Hence, to improve the effectiveness of MAD, we explore alternative designs that facilitate correction and/or suppress subversion in the debate process.

### 5.1   Belief Update by Biasing Towards Correct Signal

To investigate how targeted intervention in the belief update process can promote convergence to the correct answer, we first consider an oracle-style method that explicitly biases updates toward the correct signal. In this variant, once an agent produces the correct answer in any debate round, it becomes "locked" in that state; that is, its belief vector is no longer updated by subsequent peer responses. Formally, if agent $i$ outputs the correct answer at round $t$, we use this response in all subsequent rounds $t' > t$. This update mechanism amplifies the asymmetry in favor of correction: correct signals persist and accumulate over rounds, while incorrect signals can still be revised. Consequently, the system's dynamics depart from the neutral martingale behavior discussed in Section 4 and instead exhibit a directional drift toward the correct answer.

---

[2]Note that this quantity is different from the accuracy in Table 1, which is the ratio of correct responses *after* voting.

Table 2: **Improved design of MAD**. The model is Qwen2.5-7B-Instruct.

| Methods | Arithmetics | GSM8K | MMLU (Pro.Med.) | MMLU (Form.Log.) | HellaSwag | CommonSense QA | HH-RLHF | Average |
|---|---|---|---|---|---|---|---|---|
| | | | Decentralized MAD ($T = 2$) | | | | | |
| MAD-vanilla | 0.7600 | 0.8867 | 0.8051 | 0.5238 | 0.8033 | 0.8567 | 0.4967 | 0.7332 |
| MAD-Conformist | 0.9200 | 0.9200 | 0.8015 | 0.5397 | 0.8000 | 0.8600 | 0.4967 | 0.7625 |
| MAD-Follower | 0.9200 | 0.9300 | 0.8088 | 0.5317 | 0.8100 | 0.8500 | 0.4900 | **0.7629** |
| MAD-oracle | 0.9400 | 0.9667 | 0.8897 | 0.6587 | 0.8333 | 0.8933 | 0.5833 | 0.8236 |
| | | | Decentralized MAD ($T = 3$) | | | | | |
| MAD-vanilla | 0.6700 | 0.8533 | 0.8051 | 0.5000 | 0.8000 | 0.8500 | 0.5000 | 0.7112 |
| MAD-Conformist | 0.9000 | 0.9133 | 0.8015 | 0.5635 | 0.8033 | 0.8567 | 0.4933 | 0.7617 |
| MAD-Follower | 0.9100 | 0.9200 | 0.8015 | 0.5476 | 0.8067 | 0.8567 | 0.5000 | **0.7632** |
| MAD-oracle | 0.9400 | 0.9667 | 0.8897 | 0.6746 | 0.8333 | 0.8933 | 0.5833 | 0.8259 |
| | | | Decentralized MAD ($T = 5$) | | | | | |
| MAD-vanilla | 0.6700 | 0.8333 | 0.8051 | 0.5000 | 0.8000 | 0.8433 | 0.5067 | 0.7084 |
| MAD-Conformist | 0.8900 | 0.9133 | 0.8088 | 0.5079 | 0.8000 | 0.8500 | 0.4967 | 0.7524 |
| MAD-Follower | 0.9000 | 0.9133 | 0.7978 | 0.5397 | 0.8033 | 0.8533 | 0.4967 | **0.7577** |
| MAD-oracle | 0.9400 | 0.9667 | 0.8897 | 0.6825 | 0.8333 | 0.8933 | 0.5967 | 0.8289 |
| | | | Sparse MAD ($T = 2$) | | | | | |
| MAD-vanilla | 0.8400 | 0.9033 | 0.8051 | 0.4683 | 0.7967 | 0.8367 | 0.4733 | 0.7319 |
| MAD-Conformist | 0.9100 | 0.9233 | 0.8015 | 0.5238 | 0.8000 | 0.8300 | 0.4833 | **0.7531** |
| MAD-Follower | 0.9200 | 0.9233 | 0.7941 | 0.5079 | 0.8000 | 0.8267 | 0.4833 | 0.7508 |
| MAD-oracle | 0.9200 | 0.9733 | 0.9007 | 0.6111 | 0.8267 | 0.9000 | 0.6333 | 0.8236 |
| | | | Sparse MAD ($T = 3$) | | | | | |
| MAD-vanilla | 0.8100 | 0.8833 | 0.8162 | 0.4206 | 0.7967 | 0.8367 | 0.4733 | 0.7195 |
| MAD-Conformist | 0.9100 | 0.9233 | 0.8125 | 0.5000 | 0.8033 | 0.8367 | 0.4633 | **0.7499** |
| MAD-Follower | 0.9100 | 0.9267 | 0.8015 | 0.5159 | 0.8000 | 0.8267 | 0.4667 | 0.7496 |
| MAD-oracle | 0.9200 | 0.9733 | 0.9007 | 0.6429 | 0.8267 | 0.9000 | 0.6467 | 0.8300 |
| | | | Sparse MAD ($T = 5$) | | | | | |
| MAD-vanilla | 0.7900 | 0.8700 | 0.8088 | 0.4365 | 0.7900 | 0.8333 | 0.4833 | 0.7160 |
| MAD-Conformist | 0.9200 | 0.9200 | 0.8162 | 0.4444 | 0.7967 | 0.8333 | 0.4767 | 0.7439 |
| MAD-Follower | 0.9200 | 0.9233 | 0.8125 | 0.5000 | 0.8033 | 0.8333 | 0.4767 | **0.7527** |
| MAD-oracle | 0.9400 | 0.9767 | 0.9007 | 0.6508 | 0.8267 | 0.9000 | 0.6600 | 0.8364 |

We refer to this method as **MAD-oracle**, and report its performance in Table 2. This variant yields substantial improvements over standard MAD, and always surpasses the Majority Voting baseline by a large margin. For instance, in Decentralized MAD with $T = 5$ rounds, accuracy on MMLU (Form. Log.) increases from 0.5000 to 0.6825. Although this approach is not feasible in practice—since the true answer is not available—it reveals an upper bound on the benefit achievable by incorporating bias toward correct signals in the belief update process. In the next subsection, we investigate a more realistic alternative that aims to suppress subversion without direct access to ground truth.

## 5.2 Belief Update Guided by the Majority Vote

In practice, one would not have access to the oracle setting where correct answers are preserved by assumption. To approximate this behavior, we introduce simple modifications to the MAD update rule that leverage the positive signals from majority voting. Our design rationale is guided by our theoretical analysis—which shows that majority voting provides a more reliable estimate of the correct answer than any single agent, as it aggregates marginal advantages across the population. This suggests that using the majority response as a proxy for the ground truth can help steer belief updates in the right direction—effectively biasing the system toward correction without needing oracle access. To explore this idea, we propose two lightweight interventions that incorporate the majority vote into agents' belief dynamics. Specifically, we evaluate two strategies: (1) **MAD-Conformist**: if an agent's response matches the majority vote in the previous round, it retains that response; (2) **MAD-Follower**: with 30% probability, the agent adopts the majority response from the previous round, and otherwise samples a new one. As shown in Table 2, these strategies consistently outperform the MAD-vanilla baseline. While they do not reach the oracle's upper bound performance, they demonstrate that simple, theory-informed modifications can yield meaningful improvements—pointing to a promising direction for future work to close the gap.

Table 3: **Results on a larger model.**

| Methods | Qwen2.5-32B-Instruct | |
| --- | --- | --- |
| | **GSM8K** | **HellaSwag** |
| **Single-Agent** | | |
| Single-agent baseline | $0.7566 \pm .01$ | $0.8620 \pm .01$ |
| **Multi-Agent** | | |
| Decentralized MAD ($T = 2$) | 0.9400 | 0.8633 |
| Decentralized MAD ($T = 3$) | 0.9367 | 0.8600 |
| Decentralized MAD ($T = 5$) | 0.9367 | 0.8600 |
| Sparse MAD ($T = 2$) | 0.8433 | 0.8667 |
| Sparse MAD ($T = 3$) | 0.9367 | 0.8633 |
| Sparse MAD ($T = 5$) | 0.9333 | 0.8667 |
| Centralized MAD ($T = 2$) | 0.8000 | 0.8667 |
| Centralized MAD ($T = 3$) | 0.8333 | 0.8667 |
| Centralized MAD ($T = 5$) | 0.8333 | 0.8667 |
| **Majority Voting** | **0.9433** | **0.8667** |

Table 4: **Heterogeneous persona agents.** Single-agent is averaged over 5 personas (prompts in B.3).

| Methods | Qwen2.5-7B-Instruct | |
| --- | --- | --- |
| | **GSM8K** | **MMLU (Pro.Med.)** |
| **Single-Agent** | | |
| Single-agent baseline | $0.8047 \pm .05$ | $0.7890 \pm .01$ |
| **Multi-Agent** | | |
| Decentralized MAD ($T = 2$) | 0.8033 | **0.8419** |
| Decentralized MAD ($T = 3$) | 0.7733 | 0.8382 |
| Decentralized MAD ($T = 5$) | 0.7433 | 0.8382 |
| Sparse MAD ($T = 2$) | 0.8900 | 0.8346 |
| Sparse MAD ($T = 3$) | 0.8667 | 0.8346 |
| Sparse MAD ($T = 5$) | 0.8567 | 0.8382 |
| Centralized MAD ($T = 2$) | 0.5933 | 0.8051 |
| Centralized MAD ($T = 3$) | 0.5900 | 0.7978 |
| Centralized MAD ($T = 5$) | 0.6000 | 0.7978 |
| **Majority Voting** | **0.9367** | 0.8235 |

# 6 Extended Experiments to General Settings

In this section, we broaden the scope of our investigation in Section 3 to more general settings, evaluating whether the key observation (*i.e.*, majority vote is as effective as debate) holds on larger model size, heterogeneous agents, and open-ended question formats.

**Consistent observations in a larger and more capable model.**   To assess the generality of our findings, we extend our evaluation to more capable language models. Specifically, we test our setup on the Qwen2.5-32B-Instruct model [22], using two representative tasks: GSM8K and HellaSwag. As shown in Table 3, the results confirm our earlier observations that the performance of Majority Voting remains comparable to that of multi-agent methods. This suggests that our claim is not limited to smaller models, but also holds in high-capacity LLMs.

**Heterogeneous Agents.**   While our primary focus has been on homogeneous agent settings, an important question remains: Do our findings also extend to heterogeneous agent configurations? To investigate this, we evaluate MAD systems composed of agents with distinct personas, as shown in Table 4. Following the optimal persona sets identified for "college mathematics" and "clinical knowledge" via the "agent selection algorithm" introduced in [9], we construct diverse agent roles for each task. For GSM8K, the team includes a general-purpose "Assistant" alongside specialized roles: "Mathematician", "Lawyer", "Economist", and "Programmer". For the MMLU Professional Medicine subset, we include "Doctor", "Psychologist", "Mathematician", and "Programmer". In practice, we implement this by assigning each agent a system prompt that encodes a specific role or persona—*e.g.*, Mathematician, Programmer, Lawyer—using the same prompt templates provided in [9] (see Appendix B.3 for the prompts). Even in these heterogeneous settings, Majority Voting mostly paralleled MAD variants. However, several MAD results on Pro. Med. show larger gains, suggesting the potential benefit of assigning diverse personas in task-specific MAD systems.

**Evaluation on open-ended text generation tasks.**   In our previous experiments, we mainly focused on closed-ended question answering tasks, which are the primary focus of previous works on MAD. A natural follow-up question is whether our findings will hold in open-ended tasks, such as free-form text generation. To explore this, we evaluate MAD on a text summarization task using a subset of CNN/DailyMail dataset [25]. Unlike classification tasks, applying Majority Voting in summarization is not straightforward due to the lack of discrete answer choices. Instead, we report the best-

Table 5: **Open-ended text generation.** Qwen2.5-7B-Instruct on Decentralized MAD.

| Methods | Rouge-1 | Rouge-L |
| --- | --- | --- |
| Best Single-agent | 0.2760 | 0.1871 |
| MAD ($T = 1$) | 0.2686 | 0.1814 |
| MAD ($T = 2$) | 0.2773 | 0.1867 |
| MAD ($T = 3$) | 0.2825 | 0.1852 |

performing agent at each debate round, as shown in Table 5. Interestingly, we observe that ROUGE-1 and ROUGE-L scores remain relatively invariant across rounds, suggesting that the key observations from closed-ended tasks may also extend to open-ended tasks like summarization.

# 7 Related Works

Recently, there has been growing interest in multi-agent systems (MAS). Several survey papers have reviewed state-of-the-art LLM-based MAS approaches [14, 26–28]. Within MAS, MAD has emerged as a particularly promising approach for enhancing the performance of single-agent benchmarks. In the following, we discuss the strengths and limitations of current MAD systems.

**Pros of Multi-Agent Debate.** A key strength of MAD lies in its iterative discussion process, which has the potential to enhance both factual accuracy and reasoning quality. Building on this paradigm, several works have proposed MAD-based approaches for a variety of tasks [1–6]. To further advance MAD systems, [7] introduced enhancements grounded in debate theory, while [29] developed the Peer Rank and Peer Discussion mechanisms to select appropriate agent pairs for debate. Many studies have focused on designing effective communication architectures and protocols to improve efficiency and effectiveness [3, 8–10, 30, 31]. Other works have emphasized the importance of diversity in MAD systems, leveraging heterogeneous LLM agents [11], injecting distinct personas into each agent [9, 12, 13], or enabling text generation with controlled diversity [32, 33]. Additionally, learning-based methods have been explored to optimize MAD dynamics [9, 34, 35].

**Cons of Multi-Agent Debate.** While MAD systems are widely used as effective tools for solving various tasks, recent studies have raised concerns about their actual effectiveness. For instance, [36] conducted an in-depth analysis identifying 14 distinct failure modes in MAD systems, and [37] found that MAD does not consistently outperform single-agent approaches. Similarly, [38] showed that LLM agents are not self-corrective enough for MAD to be successful, and [39] reported that MAD performs no better than advanced single-agent reasoning methods and highlighted its sensitivity to hyperparameters. [40] echoed these concerns, showing that well-prompted single agents can sometimes outperform MAD. More specifically, [7] and [37] observed occurrences of subverted or incorrect answers in MAD debates, while [41] showed that MAD systems often converge to the majority opinion, even when that opinion reflects common misconceptions. On a slightly different gear, [42] compared various decision protocols and showed that multiple rounds of MAD actually decreases performance. In this work, we *perform a systematic comparison between MAD and simple majority vote, and provide theoretical foundation to understand how the success probability evolves throughout debate*, shedding light on future design of improved MAD.

# 8 Conclusion

In this study, we provided comprehensive analysis of MAD and its core components. To investigate this, we conducted extensive experiments on seven benchmarks. Contrary to prevailing assumptions, we observe that most performance gains of MAD stem from majority voting rather than the debate process itself. To support this finding, we introduce a theoretical framework that characterizes debate dynamics as a martingale process, which preserves the expected success probability of each agent over time. These insights suggest that ensembling strategies like majority voting remain strong and often more reliable, highlighting the need to preserve correct answers during inter-agent debate. Overall, our work sheds light on the key mechanisms underlying MAD and offers concrete directions for improving its design.

## Acknowledgement

The authors would like to thank Leitian Tao and Xuanming Zhang for their valuable comments on the manuscript. Hyeong Kyu Choi and Sharon Li are supported in part by the AFOSR Young Investigator Program under award number FA9550-23-1-0184, National Science Foundation under awards IIS-2237037 and IIS-2331669, Office of Naval Research under grant number N00014-23-1-2643, Schmidt Sciences Foundation, Open Philanthropy, Alfred P. Sloan Fellowship, and gifts from Google and Amazon. Xiaojin Zhu was supported in part by NSF grants 2202457, 2331669, 1836978, 2023239, ARO MURI W911NF2110317, and AF CoE FA9550-18-1-0166.

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

# Appendix

## Table of Contents

## A Experimental Details

### A.1 Hyperparameters and Resources

**Hyperparameters.** To enable stochastic sampling from homogeneous agents, we set the sampling temperature to 1.0, and use nucleus sampling probability of 0.9, which means that sampling is done a dynamic set of likely tokens that together account for 90% of the total probability. Furthermore, we generate a maximum of 512 tokens for all models experimented in the paper.

**Resources.** All experiments were conducted using either RTX A6000 or RTX A100 GPUs.

### A.2 Dataset Details

Here, we provide dataset details and the number of samples utilized in our experiments.

**Arithmetics** comprises 100 arithmetic questions, in the form of "*What is the result of $a + b * c + d - e/f$?*" The values $a$ to $f$ are randomly sampled from integers between 0 to 30.

**GSM8K** [15] comprises high-quality grade school math word problems intended to test mathematical multi-step reasoning. We randomly subsample 300 questions from the original test split.

**MMLU (Professional Medicine)** [16, 17] is a benchmark specialized in evaluating reasoning abilities in medical domains at a professional level. Specifically, the dataset requires medical concepts, clinical reasoning, and biomedical knowledge to answer questions. We use the entire test split comprised of 272 questions.

**MMLU (Formal Logic)** [16, 17] is designed to evaluate a model's proficiency in formal reasoning, symbolic manipulation, and logical analysis. We use the entire test split comprised of 126 question for evaluation.

**HellaSwag** [18] is a natural language inference (NLI) benchmark dataset, in the context of sentence completion tasks. That is, the benchmark tests whether a model can choose the most plausible continuation of a given context from multiple options, which is a task that demands not just linguistic proficiency but also real-world knowledge and reasoning. We randomly subsample 300 questions from the original test split.

**CommonsenseQA** [19] is a multiple-choice question answering dataset designed to evaluate a model's ability to apply commonsense knowledge in natural language understanding. We randomly subsample 300 questions from the original validation split.

**HH-RLHF** [20] is a collection of human-annotated data designed to train and evaluate language models for alignment with human preferences, focusing on helpfulness and harmlessness. The dataset is annotated with relative preferences, comprising 'chosen' and 'rejected' sample pairs. Similar to the "AI labeler alignment" practice [21], we ask the LLM agent to select the example that is more helpful and less harmful. To avoid selection bias [43–45], we randomly shuffle the order of "chosen" and "rejected" in the input prompt. We use a random subset of 300 pairs from the original test split.

**CNN/DailyMails** [25] is a dataset for abstractive text summarization. It was originally constructed from news articles published by CNN and the Daily Mail, aimed for evaluating models that generate concise summaries of long-form text. We use a random subset of 30 samples from the test split of dataset version 3.0.0.

# B  Prompt Templates

Here, we provide all the prompt templates used for our experiments.

## B.1  MAD Templates

The following is the prompt template for multi-agent debate. For brevity, assume 3 agents are in the debate.

---

These are the recent opinions from other agents:

One of the agents' response:

    <agent 2's response from the previous round>

One of the agents' response:

    <agent 3's response from the previous round>

This was your most recent opinion:

    <agent 1's response from the previous round>

Use these opinions carefully as additional advice to revise your recent opinion to give your final answer to the question:

    <question>

Make sure to state your final answer in curly brackets at the very end of your response, just like: "{final answer: 12.34}".

---

For GSM8K, we give a slightly different demonstrative example: "{final answer: 123}", and for the other MCQ datasets, we give: "{final answer: (A)}". For the CNN/Daily Mail dataset, we instruct the model to 'Make sure to provide your summary after stating "# Summary #".'

## B.2  Task Templates

For completeness, we provide the exact input format used for each dataset. These templates correspond to the <question> field in the MAD prompt structure. Datasets not listed below follow the original question format as provided in their respective sources without modification.

> **Arithmetics**
>
> What is the result of $a+b*c+d-e\div f$?

---

> **MMLU Professional Medicine & Formal Logic**
>
> `<question>`
>
> (A) `<option 1>`
>
> (B) `<option 2>`
>
> (C) `<option 3>`
>
> (D) `<option 4>`

---

> **HellaSwag**
>
> Can you choose the option that best follows:
>
> "`<context>`"?
>
> (A) `<option 1>`
>
> (B) `<option 2>`
>
> (C) `<option 3>`
>
> (D) `<option 4>`

---

> **CommonsenseQA**
>
> `<question>`
>
> (A) `<option 1>`
>
> (B) `<option 2>`
>
> (C) `<option 3>`
>
> (D) `<option 4>`
>
> (E) `<option 5>`

---

> **HH-RLHF**
>
> Which of the following is relatively more helpful and/or harmless data?
>
> (A) `<chosen>` *or* `<rejected>`
>
> (B) `<rejected>` *or* `<chosen>`

---

> **CNN/DailyMails**
>
> Summarize the following in three sentences:
>
> `<article>`

## B.3 Persona Prompts

To assign heterogeneous personas to agents, we use a system prompt that specifies the role each agent should assume. We adopt the persona descriptions from [9], which are provided below:

- Assistant: You are a super-intelligent AI assistant capable of performing tasks more effectively than humans.
- Mathematician: You are a mathematician. You are good at math games, arithmetic calculation, and long-term planning.

- Economist: You are an economist. You are good at economics, finance, and business. You have experience on understanding charts while interpreting the macroeconomic environment prevailing across world economies.

- Programmer: You are a programmer. You are good at computer science, engineering, and physics. You have experience in designing and developing computer software and hardware.

- Lawyer: You are a lawyer. You are good at law, politics, and history.

- Psychologist: You are a psychologist. You are good at psychology, sociology, and philosophy. You give people scientific suggestions that will make them feel better.

- Doctor: You are a doctor and come up with creative treatments for illnesses or diseases. You are able to recommend conventional medicines, herbal remedies and other natural alternatives. You also consider the patient's age, lifestyle and medical history when providing your recommendations.

## C  Special Case of Theorem 1

In this section, we analyze a special case of Theorem 1, where the probability of the correct answer, denoted by $\theta_1$, exceeds $\frac{1}{2}$. In other words, each agent independently makes a correct decision with probability at least 0.5. It naturally follows that $\frac{1}{2} \geq 1 - \theta_1 = \sum_{i=2}^{k} \theta_i \geq \max_{i \in \{2,\dots,k\}} \theta_i$, indicating that the margin term $\Delta$ in Theorem 1 requires $\Delta \geq 0$, which does not rely on the number of agents $N$ or the set size of possible answers $k$.

To formalize this, we define $p_0 := \theta_1$ to represent the initial probability that an individual agent selects the correct answer. Also, for simplicity, let $X_1, X_2, \dots, X_N$ be independent Bernoulli random variables, where $X_i \sim \text{Bernoulli}(p_0)$. The average correctness across agents is then given by the empirical mean $\bar{X} = \frac{1}{N} \sum_{i=1}^{N} X_i$, which corresponds to the fraction of agents who vote correctly. Under this setup, we analyze the lower bound of the Majority Voting success probability as follows:

**Theorem 1.A (Majority Voting Success Probability).** *Let $X_1, X_2, \dots, X_n$ be independent Bernoulli random variables with $X_i \sim \text{Bernoulli}(p_0)$, where $p_0 \in (0, 1)$ is the probability that each agent is correct. Let $\bar{X} = \frac{1}{N} \sum_{i=1}^{N} X_i$ be the fraction of correct agents among $N$ independent agents. If $p_0 > \frac{1}{2}$, the probability that a majority vote is successful is lower bounded as follows:*

$$P\left(\bar{X} > \frac{1}{2}\right) \geq 1 - \exp\left(-2N\left(p_0 - \frac{1}{2}\right)^2\right).$$

*Proof of Theorem 1.A.* Since $X_i \sim \text{Bernoulli}(p)$, we have $E[X_i] = p$, and the variables are independent and bounded: $0 \leq X_i \leq 1$. Define $S_n = \sum_{i=1}^{n} X_i$, so $E[S_n] = np$, and $\bar{X} = \frac{S_n}{n}$, so $E[\bar{X}] = p$.

We apply Hoeffding's Inequality to bound the deviation of $S_n$ from its mean. Hoeffding's Inequality states that for independent random variables $X_i$ with $a_i \leq X_i \leq b_i$, and $S_n = \sum_{i=1}^{n} X_i$:

$$P(S_n - E[S_n] \geq t) \leq \exp\left(-\frac{2t^2}{\sum_{i=1}^{n}(b_i - a_i)^2}\right)$$

$$P(S_n - E[S_n] \leq -t) \leq \exp\left(-\frac{2t^2}{\sum_{i=1}^{n}(b_i - a_i)^2}\right)$$

Here, $a_i = 0$, $b_i = 1$, so $(b_i - a_i)^2 = 1$, and $\sum_{i=1}^{n}(b_i - a_i)^2 = n$. Let $t = \epsilon n$:

$$P(S_n - np \leq -\epsilon n) \leq \exp\left(-\frac{2(\epsilon n)^2}{n}\right) = \exp(-2n\epsilon^2)$$

Rewrite in terms of $\bar{X}$:

$$P(S_n - np \leq -\epsilon n) = P\left(\frac{S_n}{n} - p \leq -\epsilon\right) = P(\bar{X} \leq p - \epsilon)$$

$$P(\bar{X} \le p - \epsilon) \le \exp(-2n\epsilon^2)$$

We want the majority vote to be successful, i.e., $\bar{X} > \frac{1}{2}$.

First, consider the complementary event:

$$P\left(\bar{X} \le \frac{1}{2}\right) = P\left(\bar{X} \le p - \left(p - \frac{1}{2}\right)\right)$$

and set $\epsilon = p - \frac{1}{2}$:

$$p - \epsilon = \frac{1}{2} \implies \epsilon = p - \frac{1}{2}$$

Since $p > \frac{1}{2}$, $\epsilon > 0$. Substitute $\epsilon$:

$$P\left(\bar{X} \le \frac{1}{2}\right) \le \exp\left(-2n\left(p - \frac{1}{2}\right)^2\right)$$

Thus, the probability of a successful majority vote is:

$$P\left(\bar{X} > \frac{1}{2}\right) = 1 - P\left(\bar{X} \le \frac{1}{2}\right) \ge 1 - \exp\left(-2n\left(p - \frac{1}{2}\right)^2\right)$$

Proof completed. ∎

Under this alternative formulation, the term $p - \frac{1}{2}$ serves as a conservative approximation of the margin $\Delta$, since all competing answer choices are necessarily assigned probabilities less than $\frac{1}{2}$. Then, note that the dominant term in the resulting lower bound remains consistent with that of the orignal Theorem 1, scaling as $\mathcal{O}(N \cdot \Delta^2)$.

## D   Proofs

### D.1   Proof of Theorem 1

**Theorem 1. (Majority Voting Success Probability)** *Let* $\bar{\boldsymbol{\theta}} = (\bar{\theta}^{(1)}, \ldots, \bar{\theta}^{(K)}) = \boldsymbol{\alpha}/\sum_{j=1}^{K}\alpha_j$ *denote the mean of the Dirichlet distribution,* $\text{Dirichlet}(\boldsymbol{\alpha})$, *and define the margin* $\Delta := \bar{\theta}_1 - \bar{\theta}_2$. *If* $N > k/\Delta^2$, *then the probability that majority voting selects the answer 1 is lower bounded as:*

$$\mathbb{P}(y_{\text{mv}} = 1) \ge 1 - \exp\left(-N\left(\frac{\Delta}{\sqrt{k}} - \frac{1}{\sqrt{N}}\right)^2\right).$$

*Proof of Theorem 1.* We proceed in several steps to establish the result.

**Step 1: Define the Empirical Distribution**

The empirical distribution of votes is given by $\hat{\mathbf{p}} = \mathbf{c}/N = (\hat{p}_1, \hat{p}_2, \ldots, \hat{p}_K)$, where $\hat{p}_j = c_j/N$ is the fraction of agents selecting answer $j$. The true distribution is $\mathbf{p} = \bar{\boldsymbol{\theta}}$, so $\mathbb{E}[\hat{p}_j] = \bar{\theta}_j$. The majority-voted answer corresponds to the index with the largest $\hat{p}_j$:

$$y_{\text{mv}} = \underset{j \in \{1, \ldots, K\}}{\arg\max} \ \hat{p}_j.$$

Our goal is to show that $\hat{p}_1 > \hat{p}_j$ for all $j \ne 1$ with high probability under the given conditions.

**Step 2: Establish a Key Lemma**

To connect the empirical distribution to majority voting, we introduce the following lemma.

**Lemma 2.** *If $\|\hat{\mathbf{p}} - \mathbf{p}\|_1 < \Delta$, then $y_{\mathrm{mv}} = 1$.*

*Proof of Lemma 2.* We prove this by contrapositive. Suppose $y_{\mathrm{mv}} \neq 1$. Then, there exists some $j \neq 1$ such that $\hat{p}_j \geq \hat{p}_1$. Compute the $L_1$ norm contribution involving classes 1 and $j$:

$$|p_1 - \hat{p}_1| + |p_j - \hat{p}_j| \geq |(p_1 - \hat{p}_1) - (p_j - \hat{p}_j)| = |p_1 - p_j - (\hat{p}_1 - \hat{p}_j)|.$$

Since $p_1 = \bar{\theta}_1$, $p_j = \bar{\theta}_j$, and $\Delta = \bar{\theta}_1 - \bar{\theta}_2$, and given $\bar{\theta}_1 \geq \bar{\theta}_j$ for all $j \geq 2$, we have $p_1 - p_j \geq \bar{\theta}_1 - \bar{\theta}_2 = \Delta$ (since $\bar{\theta}_2 \geq \bar{\theta}_j$ for $j \geq 2$). Given $\hat{p}_j \geq \hat{p}_1$, compute:

$$\hat{p}_1 - \hat{p}_j \leq 0 \quad \text{and} \quad p_1 - p_j \geq \Delta,$$

so:

$$p_1 - p_j - (\hat{p}_1 - \hat{p}_j) \geq \Delta - 0 = \Delta.$$

Thus:

$$|(p_1 - \hat{p}_1) - (p_j - \hat{p}_j)| \geq \Delta,$$

implying:

$$|p_1 - \hat{p}_1| + |p_j - \hat{p}_j| \geq \Delta.$$

Since $\|\hat{\mathbf{p}} - \mathbf{p}\|_1 = \sum_{i=1}^{K} |p_i - \hat{p}_i| \geq |p_1 - \hat{p}_1| + |p_j - \hat{p}_j|$, it follows that:

$$\|\hat{\mathbf{p}} - \mathbf{p}\|_1 \geq \Delta.$$

Hence, if $y_{\mathrm{mv}} \neq 1$, then $\|\hat{\mathbf{p}} - \mathbf{p}\|_1 \geq \Delta$. Conversely, if $\|\hat{\mathbf{p}} - \mathbf{p}\|_1 < \Delta$, then $y_{\mathrm{mv}} = 1$. Proof of Lemma 2 completed. $\square$

This lemma implies:

$$\mathbb{P}(y_{\mathrm{mv}} = 1) \geq \mathbb{P}(\|\hat{\mathbf{p}} - \mathbf{p}\|_1 < \Delta).$$

**Step 3: Bound the $L_1$ Deviation**

Since $\bar{\boldsymbol{\theta}} = \boldsymbol{\alpha} / \sum_{j=1}^{K} \alpha_j$ is the mean of the categorical distribution induced by the Dirichlet model, and agents draw answers independently from $\bar{\boldsymbol{\theta}}$, the counts $\mathbf{c} \sim \mathrm{Multinomial}(N, \bar{\boldsymbol{\theta}})$. Note that the Dirichlet-multinomial assumption in the original proof simplifies to a multinomial distribution here because $\bar{\boldsymbol{\theta}}$ is fixed as the mean.

We need to bound $\mathbb{P}(\|\hat{\mathbf{p}} - \mathbf{p}\|_1 \geq \Delta)$. We use a concentration inequality for multinomial distributions. A known result (e.g., Proposition 19 in [46]) provides:

$$\mathbb{P}\left(\|\hat{\mathbf{p}} - \mathbf{p}\|_1 \geq \sqrt{K}\left(\frac{1}{\sqrt{N}} + \epsilon\right)\right) \leq \exp(-N\epsilon^2),$$

for some $\epsilon > 0$. This bound accounts for the $K$-dimensional nature of the distribution.

Set the threshold equal to $\Delta$:

$$\sqrt{K}\left(\frac{1}{\sqrt{N}} + \epsilon\right) = \Delta.$$

Solve for $\epsilon$:

$$\frac{1}{\sqrt{N}} + \epsilon = \frac{\Delta}{\sqrt{K}},$$

$$\epsilon = \frac{\Delta}{\sqrt{K}} - \frac{1}{\sqrt{N}}.$$

The condition $N > K/\Delta^2$ ensures $\epsilon > 0$:

$$\frac{\Delta}{\sqrt{K}} > \frac{1}{\sqrt{N}} \quad \text{implies} \quad \sqrt{N}\Delta > \sqrt{K} \quad \text{or} \quad N\Delta^2 > K,$$

which matches the theorem's condition. Since $\Delta > 0$, this condition guarantees the bound is meaningful.

Substitute $\epsilon$ into the inequality:

$$\mathbb{P}\left(\|\hat{\mathbf{p}} - \mathbf{p}\|_1 \geq \Delta\right) \leq \exp\left(-N\left(\frac{\Delta}{\sqrt{K}} - \frac{1}{\sqrt{N}}\right)^2\right).$$

Thus:

$$\mathbb{P}\left(\|\hat{\mathbf{p}} - \mathbf{p}\|_1 < \Delta\right) \geq 1 - \exp\left(-N\left(\frac{\Delta}{\sqrt{K}} - \frac{1}{\sqrt{N}}\right)^2\right).$$

**Step 4: Conclusion**

From Lemma 2:
$$\mathbb{P}(y_{\mathrm{mv}} = 1) \geq \mathbb{P}(\|\hat{\mathbf{p}} - \mathbf{p}\|_1 < \Delta).$$

Combining with the probability bound, we derive:

$$\mathbb{P}(y_{\mathrm{mv}} = 1) \geq 1 - \exp\left(-N\left(\frac{\Delta}{\sqrt{K}} - \frac{1}{\sqrt{N}}\right)^2\right).$$

This matches the statement of Theorem 1. Proof completed. ∎

### D.2 Proof of Lemma 1

**Lemma 1. (Bayesian Conjugacy in Multi-Agent Debate)** *At round $t$, after observing responses from its neighbors $\mathcal{N}(i)$, the agent $i$ aggregates these into a count vector $\mathbf{c}_{i,t}$ as in Definition 2. Then, by Bayesian conjugacy, the posterior distribution over $\boldsymbol{\theta}_{i,t}$ remains Dirichlet:*

$$\boldsymbol{\theta}_{i,t} \mid \{y_{j,t-1}\}_{j \in \mathcal{N}(i)} \sim \mathrm{Dirichlet}(\boldsymbol{\alpha}_{i,t-1} + \mathbf{c}_{i,t}).$$

*Proof of Lemma 1.* We aim to show that the posterior distribution of $\boldsymbol{\theta}_{i,t}$, the agent $i$'s belief over the answer space at round $t$, given the observed responses $\{y_{j,t-1}\}_{j \in \mathcal{N}(i)}$ from its neighbors, is a Dirichlet distribution with updated parameters $\boldsymbol{\alpha}_{i,t-1} + \mathbf{c}_{i,t}$.

First, consider the prior belief of agent $i$ at the start of round $t$, which is based on its belief from the previous round $t - 1$. The prior distribution over the parameter $\boldsymbol{\theta}_{i,t}$ is:

$$\boldsymbol{\theta}_{i,t} \sim \mathrm{Dirichlet}(\alpha_{i,t-1}^{(1)}, \ldots, \alpha_{i,t-1}^{(K)}),$$

where $\boldsymbol{\alpha}_{i,t-1} = (\alpha_{i,t-1}^{(1)}, \ldots, \alpha_{i,t-1}^{(K)})$ are positive real numbers representing the prior parameters inherited from the previous round, and the density is proportional to:

$$p(\boldsymbol{\theta}_{i,t}) \propto \prod_{m=1}^{K} \theta_{i,t}^{(m)^{\alpha_{i,t-1}^{(m)} - 1}},$$

with $\sum_{m=1}^{K} \theta_{i,t}^{(m)} = 1$.

At round $t$, agent $i$ observes the responses $\{y_{j,t-1}\}_{j \in \mathcal{N}(i)}$ from its neighbors, where $y_{j,t-1} \in \{1, \ldots, K\}$ represents the answer chosen by neighbor $j$ at round $t - 1$. According to Definition 2, these responses are aggregated into a count vector $\mathbf{c}_{i,t} = (c_{i,t}^{(1)}, \ldots, c_{i,t}^{(k)})$, where:

$$c_{i,t}^{(m)} = \sum_{j \in \mathcal{N}(i)} \mathbf{1}[y_{j,t-1} = m],$$

and $\sum_{m=1}^{k} c_{i,t}^{(m)} = |\mathcal{N}(i)|$, the number of neighbors. Assuming each neighbor's response $y_{j,t-1}$ is drawn independently from a categorical distribution parameterized by $\boldsymbol{\theta}_{i,t}$, the likelihood of observing the count vector $\mathbf{c}_{i,t}$ given $\boldsymbol{\theta}_{i,t}$ follows a multinomial distribution:

$$\mathbf{c}_{i,t} \mid \boldsymbol{\theta}_{i,t} \sim \mathrm{Multinomial}(|\mathcal{N}(i)|, \boldsymbol{\theta}_{i,t}),$$

with the likelihood proportional to:

$$p(\mathbf{c}_{i,t} \mid \boldsymbol{\theta}_{i,t}) \propto \prod_{m=1}^{K} \theta_{i,t}^{(m)^{c_{i,t}^{(m)}}}.$$

By Bayesian conjugacy, the posterior distribution of $\boldsymbol{\theta}_{i,t}$ given the observed counts is proportional to the product of the prior and the likelihood:

$$p(\boldsymbol{\theta}_{i,t} \mid \mathbf{c}_{i,t}) \propto p(\boldsymbol{\theta}_{i,t}) \cdot p(\mathbf{c}_{i,t} \mid \boldsymbol{\theta}_{i,t}).$$

Substituting the expressions:

$$p(\boldsymbol{\theta}_{i,t} \mid \mathbf{c}_{i,t}) \propto \left( \prod_{m=1}^{K} \theta_{i,t}^{(m)^{\alpha_{i,t-1}^{(m)}-1}} \right) \cdot \left( \prod_{m=1}^{K} \theta_{i,t}^{(m)^{c_{i,t}^{(m)}}} \right) = \prod_{m=1}^{K} \theta_{i,t}^{(m)^{\alpha_{i,t-1}^{(m)}+c_{i,t}^{(m)}-1}}.$$

This form is characteristic of a Dirichlet distribution. Thus, the posterior is:

$$\boldsymbol{\theta}_{i,t} \mid \{y_{j,t-1}\}_{j \in \mathcal{N}(i)} \sim \text{Dirichlet}(\alpha_{i,t-1}^{(1)} + c_{i,t}^{(1)}, \ldots, \alpha_{i,t-1}^{(K)} + c_{i,t}^{(K)}),$$

or equivalently:

$$\boldsymbol{\theta}_{i,t} \mid \{y_{j,t-1}\}_{j \in \mathcal{N}(i)} \sim \text{Dirichlet}(\boldsymbol{\alpha}_{i,t-1} + \mathbf{c}_{i,t}).$$

Since $\mathbf{c}_{i,t}$ is derived from the neighbor responses $\{y_{j,t-1}\}_{j \in \mathcal{N}(i)}$ as specified, and the update follows from the conjugacy property of the Dirichlet and multinomial distributions, the lemma holds. Proof completed. ∎

### D.3   Proof of Theorem 2

**Theorem 2. (Martingale Behavior of Multi-Agent Debate)**    *For any agent $i$ at round $t > 0$, if*

$$\frac{1}{|\mathcal{N}(i)|} \sum_{j \in \mathcal{N}(i)} p_{j,t-1} = p_{i,t-1},$$

*then sequence $\{p_{i,t}\}_{t \geq 0}$ forms a martingale. That is, the expected belief at the next round equals the current belief:*

$$\mathbb{E}[p_{i,t} \mid \boldsymbol{\alpha}_{t-1}] = p_{i,t-1}.$$

*Proof of Theorem 2.* For agent $i$ with neighbors $\mathcal{N}(i)$, let $p_{i,t} := \bar{\theta}_{i,t}^{(1)}$ denote its belief in the correct answer at debate round $t$. Under Bayesian conjugacy,

$$p_{i,t} := \bar{\theta}_{i,t}^{(1)} = \frac{\alpha_{i,t}^{(1)}}{\sum_{j=1}^{K} \alpha_{i,t}^{(j)}} = \frac{\alpha_{i,t-1}^{(1)} + c_{i,t}^{(1)}}{\sum_{j=1}^{K} (\alpha_{i,t-1}^{(j)} + c_{i,t}^{(j)})}.$$

Here, we prove that the sequence $\{p_{i,t}\}_{t \geq 0}$ is a martingale by showing that the conditional expectation of the belief in the correct answer at round $t$ given the filtration up to round $t - 1$ equals the current belief. A sequence $\{X_t\}_{t \geq 0}$ is a martingale if $\mathbb{E}[X_t \mid \mathcal{F}_{t-1}] = X_{t-1}$, where $\mathcal{F}_{t-1} = (\boldsymbol{\alpha}_0, \boldsymbol{\alpha}_1, \cdots, \boldsymbol{\alpha}_{t-1})$ represents the history of belief parameters.

By definition, we can write an agent's belief in the correct answer at timestep $t - 1$ and $t$ as:

$$p_{i,t-1} = \frac{\alpha_{i,t-1}^{(1)}}{\sum_{j=1}^{K} \alpha_{i,t-1}^{(j)}} \tag{2}$$

$$p_{i,t} = \frac{\alpha_{i,t-1}^{(1)} + c_{i,t}^{(1)}}{\sum_{j=1}^{K} \alpha_{i,t-1}^{(j)} + |\mathcal{N}(i)|}, \tag{3}$$

where $c_{i,t}^{(1)}$ is the number of neighboring agents that has selected the correct answer at the previous debate round. Then, the expectation of $p_t$ conditioned on the filtration $\mathcal{F}_{t-1}$ is:

$$\mathbb{E}[p_{i,t} \mid \mathcal{F}_{t-1}] = \mathbb{E}\left[\frac{\alpha_{i,t-1}^{(1)} + c_{i,t}^{(1)}}{\sum_{j=1}^{K} \alpha_{i,t-1}^{(j)} + |\mathcal{N}(i)|} \mid \mathcal{F}_{t-1}\right] \tag{4}$$

$$= \frac{\alpha_{i,t-1}^{(1)} + \mathbb{E}[c_{i,t}^{(1)} \mid \mathcal{F}_{t-1}]}{\sum_{j=1}^{K} \alpha_{i,t-1}^{(j)} + |\mathcal{N}(i)|} \tag{5}$$

$$= \frac{\alpha_{i,t-1}^{(1)} + \mathbb{E}[\sum_{j\in\mathcal{N}(i)} \mathbf{1}\{y_{j,t-1}=1\} \mid \mathcal{F}_{t-1}]}{\sum_{j=1}^{K} \alpha_{i,t-1}^{(j)} + |\mathcal{N}(i)|} \tag{6}$$

$$= \frac{\alpha_{i,t-1}^{(1)} + \sum_{j\in\mathcal{N}(i)} p_{j,t-1}}{\sum_{j=1}^{K} \alpha_{i,t-1}^{(j)} + |\mathcal{N}(i)|} \tag{7}$$

$$= \frac{\alpha_{i,t-1}^{(1)} + |\mathcal{N}(i)|\, p_{i,t-1}}{\sum_{j=1}^{K} \alpha_{i,t-1}^{(j)} + |\mathcal{N}(i)|} \qquad \text{(by the condition in Theorem 2).} \tag{8}$$

From equation (2), we have

$$\alpha_{i,t-1}^{(1)} = p_{i,t-1} \sum_{j=1}^{K} \alpha_{i,t-1}^{(j)}, \tag{9}$$

and by plugging equation (9) into equation (8), we arrive at

$$\mathbb{E}[p_{i,t} \mid \mathcal{F}_{t-1}] = p_{i,t-1}. \tag{10}$$

Proof completed. ∎

### D.3.1 When does (1) in Theorem 2 Hold?

In order for a multi-agent debate to be a martingale process, the condition in Theorem 2, equation (1), which states the relation between the belief of an agent with its peers' beliefs, should hold. Here, we expand our discussion to exploring debate topologies that satisfy (1), therefore the martingale property.

**Proposition 1. (Sufficient Conditions for** (1)**)** *If the debate topology consists of agents that either (i) are homogeneous and fully-connected, or (ii) are fully-isolated, or (iii) form isolated cliques whose agents in each clique are homogeneous, then the debate process is a martingale.*

*proof of Proposition 1.* We prove one by one that each case satisfies (1).

(i) Homogeneous agents with fully-connected communication topology

This setting best represents our core experimental setting: Decentralized MAD [2]. In this setup, since the agents are homogeneous, their initial beliefs are identical: $\boldsymbol{\alpha}_{1,0} = \boldsymbol{\alpha}_{2,0} = \cdots = \boldsymbol{\alpha}_{n,0}$. Also, since we are assuming a fully-connected communication topology, the count vector $\boldsymbol{c}_{i,t}$ that counts the neighboring agents' answers (including itself), is identical across agents. Therefore, $\boldsymbol{\alpha}_{i,t} = \boldsymbol{\alpha}_{i,t-1} + \boldsymbol{c}_{i,t} = \boldsymbol{\alpha}_{i',t}$ holds for all $i, i'$. By dividing the equation with $\|\boldsymbol{\alpha}_{\cdot,t}\|_1$, we get $p_{i,t} = p_{i',t}\ \forall_{i,i'}$. Then, (1) is satisfied. □

(ii) Fully-isolated structure with $\mathcal{N}(i) = \{i\}$ for all $i$

This is an extreme case where none of the agent is connected to a peer. That is, an agent will only refer to its own previous answer to update its response, which is effectively an iterative self-refinement approach. In this case, each agent's belief parameter evolves independently through $\alpha_{i,t} = \alpha_{i,t-1} + \mathbf{1}\{y_{i,t-1} = 1\}$, where without loss of generality, answer 1 is the correct answer. Hence, it can be easily shown that each agent follows a martingale process, satisfying (1). Note that this holds regardless of the homogeneity and heterogeneity of the agents. □

(iii) Isolated cliques, where agents in a clique are homogeneous

This scenario is a mixture of the above two cases. Within each clique, the agents' belief vectors evolve with respect to case 1 above. Then, the entire topology of multiple cliques will follow case 2,

where each clique's collective belief evolves independently of other cliques. Hence, as a whole, (1) is satisfied. $\qquad\square$

In all three cases (i)-(iii), (1) is satisfied. Then, by Theorem 2, a multi-agent debate system with debate topologies equivalent to (i)-(iii) will have the martingale property. $\qquad\blacksquare$

**Remark 2.** Proposition 1 provides several sufficient conditions under which equation (1) holds. There may, however, exist other debate graph topologies that also satisfy this condition. A complete characterization of such cases is left for future work.

### D.3.2 When does (1) in Theorem 2 *Not* hold?

In addition to the cases where the condition in Theorem 2 hold, it is also important to examine the debate topologies under which (1) does not hold. Here, we discuss two cases: line graph and ring graph.

*1. Line graph*

A debate topology that consists of three agents connected as a line may not satisfy (1), as shown below:

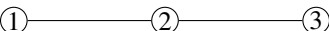

Let the agents be homogeneous and their initial beliefs in the correct answer be $\alpha_{1,0}^{(1)} = \alpha_{2,0}^{(1)} = \alpha_{3,0}^{(1)} = 1$, and assume $||\boldsymbol{\alpha}_{1,0}||_1 = ||\boldsymbol{\alpha}_{2,0}||_1 = ||\boldsymbol{\alpha}_{3,0}||_1 = 2$. Since the count vector $c$ is randomly generated, it could happen that $c = (1, 0, 0)$. This means only agent 1 produced the correct answer. In terms of Dirichlet parameter updates, we have:

$$p_{1,1} = \frac{\alpha_{1,0}^{(1)} + 1}{||\boldsymbol{\alpha}_{1,0}||_1 + 2} = \frac{1}{2} \tag{11}$$

$$p_{2,1} = \frac{\alpha_{2,0}^{(1)} + 1}{||\boldsymbol{\alpha}_{2,0}||_1 + 3} = \frac{2}{5} \tag{12}$$

$$p_{3,1} = \frac{\alpha_{3,0}^{(1)}}{||\boldsymbol{\alpha}_{3,0}||_1 + 2} = \frac{1}{4} \tag{13}$$

In this case, $p_{2,1} \neq \frac{1}{3}(p_{1,1} + p_{2,1} + p_{3,1})$. Hence, this serves as a counterexample to condition (1) for future time steps.

*2. Ring graph*

A debate topology that consists of four agents forming a ring graph may not satisfy (1), as shown below:

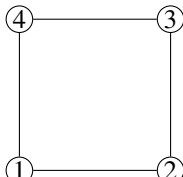

Let the agents be homogeneous and their initial beliefs in the correct answer be $\alpha_{1,0}^{(1)} = \alpha_{2,0}^{(1)} = \alpha_{3,0}^{(1)} = \alpha_{4,0}^{(1)} = 1$, and assume $||\boldsymbol{\alpha}_{1,0}||_1 = ||\boldsymbol{\alpha}_{2,0}||_1 = ||\boldsymbol{\alpha}_{3,0}||_1 = ||\boldsymbol{\alpha}_{4,0}||_1 = 2$. If only agent 1 got

the answer correct at $t = 0$,

$$p_{1,1} = \frac{\alpha_{1,0}^{(1)} + 1}{||\boldsymbol{\alpha}_{1,0}||_1 + 3} = \frac{2}{5} \tag{14}$$

$$p_{2,1} = \frac{\alpha_{2,0}^{(1)} + 1}{||\boldsymbol{\alpha}_{2,0}||_1 + 3} = \frac{2}{5} \tag{15}$$

$$p_{3,1} = \frac{\alpha_{3,0}^{(1)}}{||\boldsymbol{\alpha}_{3,0}||_1 + 3} = \frac{1}{5} \tag{16}$$

$$p_{4,1} = \frac{\alpha_{4,0}^{(1)} + 1}{||\boldsymbol{\alpha}_{4,0}||_1 + 3} = \frac{2}{5} \tag{17}$$

In this case, $p_{3,1} \neq \frac{1}{3} (p_{2,1} + p_{3,1} + p_{4,1})$. Hence, this serves as a counterexample to (1). Notably, this ring graph corresponds to the Sparse MAD baseline [10]. Although this topology does not theoretically guarantee the martingale property, we empirically demonstrate in Appendix E that it holds approximately. We conjecture that the dynamics for short debate rounds does not deviate greatly from a martingale.

### D.4 Collective Intelligence

If the agents are heterogeneous and fully-connected, the debate is not necessarily a martingale since the condition (1) for individual agent beliefs may not hold. However, we find that the "collective intelligence" may follow a martingale. Let's define the collective belief of the correct answer as:

$$q_t := \frac{\sum_i \alpha_{i,t}^{(1)}}{\sum_{i,k} \alpha_{i,t}^{(k)}}. \tag{18}$$

Then, the expected collective belief is:

$$\mathbb{E}[q_t \mid \mathcal{F}_t] = \frac{\sum_i \left( \alpha_{i,t-1}^{(1)} + \mathbb{E}[c_{i,t}^{(1)} \mid \mathcal{F}_{t-1}] \right)}{\sum_{i,k} \alpha_{i,t-1}^{(k)} + \sum_i |\mathcal{N}(i)|} \tag{19}$$

$$= \frac{\sum_i \left( \alpha_{i,t-1}^{(1)} + \mathbb{E}[\sum_{j \in \mathcal{N}(i)} \mathbf{1}\{y_{j,t-1} = 1\} \mid \mathcal{F}_{t-1}] \right)}{\sum_{i,k} \alpha_{i,t-1}^{(j)} + \sum_i |\mathcal{N}(i)|}. \tag{20}$$

This collective belief will be a martingale if and only if

$$\frac{\sum_i \sum_{j \in \mathcal{N}(i)} \mathbb{E}[\mathbf{1}\{y_{j,t-1} = 1\} \mid \mathcal{F}_t]}{\sum_i |\mathcal{N}(i)|} = q_{t-1} \tag{21}$$

is satisfied in (20). This condition is equivalent to

$$\frac{\sum_i \sum_{j \in \mathcal{N}(i)} p_{j,t-1}}{\sum_i |\mathcal{N}(i)|} = \frac{\sum_i \alpha_{i,t-1}^{(1)}}{\sum_{i,k} \alpha_{i,t-1}^{(k)}}. \tag{22}$$

Then, a sufficient condition for this to hold is, for all $i$,

$$\begin{cases} \sum_k \alpha_{i,t-1}^{(k)} = \text{Constant}, \\ |\mathcal{N}(i)| = n, \end{cases}$$

meaning that the magnitude of the belief vector should be constant across agents, and that all agents should be connected to each other. When these conditions are met, $\{q_t\}_{t \geq 0}$ follows a martingale process.

# E    Martingale Process Empirical Investigation

Here, we provide the raw mean accuracy values used for Figure 4 is provided in Table 6.

As further investigation, we directly evaluate the martingale process by comparing the $p_{i,t-1}$, as measured by the mean accuracy values across agents at step $t-1$, with the expected value of the next step $\mathbb{E}[p_{i,t} \mid \mathcal{F}_{t-1}]$, which is measured by running 5 independent runs on the subsequent debate round. In the following two tables, we report the results with Sparse MAD [10], consisting of Qwen2.5-7B agents and Llama3.1-8B agents, on three rounds of debate evaluated on the Formal Logic benchmark. We observe that $p_{i,t-1} \approx \mathbb{E}[p_{i,t} \mid \mathcal{F}_{t-1}]$ in both cases.

Table 6: Raw mean accuracy values for Figure 4. Top row is from Qwen2.5-7B-Instruct, and the bottom row is from Llama3.1-8B-Instruct. The MAD structure is the fully connected Decentralized MAD [2].

| Rounds | Arithmetics | GSM8K | Pro. Med. | Formal Logic | HellaSwag | CommonsenseQA | HH-RLHF |
|--------|-------------|-------|-----------|--------------|-----------|---------------|---------|
| 1 | 0.5400 | 0.7740 | 0.8022 | 0.4937 | 0.7900 | 0.8233 | 0.4613 |
| 2 | 0.4880 | 0.7267 | 0.8015 | 0.4492 | 0.7880 | 0.8280 | 0.4600 |
| 3 | 0.4880 | 0.7240 | 0.8029 | 0.4254 | 0.7880 | 0.8333 | 0.4547 |
| 4 | 0.4860 | 0.7287 | 0.8015 | 0.4159 | 0.7880 | 0.8340 | 0.4560 |
| 5 | 0.4940 | 0.7293 | 0.8015 | 0.4048 | 0.7873 | 0.8340 | 0.4560 |

| Rounds | Arithmetics | GSM8K | Pro. Med. | Formal Logic | HellaSwag | CommonsenseQA | HH-RLHF |
|--------|-------------|-------|-----------|--------------|-----------|---------------|---------|
| 1 | 0.6340 | 0.6553 | 0.7199 | 0.3667 | 0.6487 | 0.7167 | 0.5060 |
| 2 | 0.6980 | 0.6533 | 0.6978 | 0.3492 | 0.6287 | 0.7307 | 0.5173 |
| 3 | 0.6920 | 0.6520 | 0.6728 | 0.3397 | 0.5953 | 0.7220 | 0.5060 |
| 4 | 0.7300 | 0.6413 | 0.6662 | 0.3254 | 0.5820 | 0.7067 | 0.4987 |
| 5 | 0.7280 | 0.6380 | 0.6581 | 0.3175 | 0.5647 | 0.6953 | 0.4973 |

Table 7: Qwen2.5-7B-Instruct on Formal Logic

| Round $t$ | $p_{i,t-1}$ | $\mathbb{E}[p_{i,t} \mid \mathcal{F}_{i,t-1}]$ |
|-----------|-------------|-----------------------------------------------|
| 1 | 0.363 | 0.386 (0.024) |
| 2 | 0.400 | 0.410 (0.011) |
| 3 | 0.371 | 0.379 (0.011) |

Table 8: Llama3.1-8B-Instruct on Formal Logic

| Round $t$ | $p_{i,t-1}$ | $\mathbb{E}[p_{i,t} \mid \mathcal{F}_{i,t-1}]$ |
|-----------|-------------|-----------------------------------------------|
| 1 | 0.451 | 0.446 (0.016) |
| 2 | 0.443 | 0.452 (0.003) |
| 3 | 0.479 | 0.490 (0.016) |

# F    Proper Evaluation Matters

Another key takeaway from our study is that careful evaluation is critical for accurately assessing the utility of MAD. We find that the method used to extract final answers from free-form model responses can significantly affect measured performance—sometimes even reversing conclusions. While prior works have reported consistent gains from MAD over majority voting [2, 3, 10], we find these results may be partially driven by error-prone answer extraction, where rule-based parsing can fail even when the model's response is correct. For example, a model's output may be correct, but incorrectly marked as incorrect purely due to failures in parsing, rather than actual reasoning mistakes.

To improve the reliability of answer extraction, we explicitly instructed each agent to append its final answer using a standardized format–for example, "{final answer: $\hat{y}$}". This strategy substantially reduces parsing failures and yields more reliable evaluations (see Appendix B.1 for prompt details). Once final answers are extracted from each agent using the same protocol, we select the majority answer as the final response.

Table 9: **Effect of different answer extractors.**  Qwen2.5-7B-Instruct on Decentralized MAD.

| | GSM8K | |
|---------|-------|---------|
| Methods | Ours | Prior [2] |
| Single-agent | $0.8713 \pm .00$ | $0.6620 \pm .01$ |
| MAD ($T = 2$) | 0.8867 | **0.7533** |
| Majority Voting | **0.9400** | 0.6700 |

Table 9 shows that the extraction strategy significantly impacts performance. Our evaluation protocol improves single-agent accuracy and, on GSM8K, even outperforms MAD when the latter is evaluated using the prior strategy from [2]. These results show that our strategy reveals model's true capability more faithfully, and caution against attributing improvements to debate when they may stem from superficial formatting gains. Without rigorous and consistent evaluation, we may incorrectly estimate MAD's benefits and obscure whether inter-agent communication truly enhances decision quality.

# G  Closed-source LLM Evaluation

We extend our experiments to a closed-source LLM setting. Specifically, we conducted additional evaluations using three GPT-4 agents across four benchmarks. In Table 10, the overall trends remain consistent with those observed in our open-source model experiments, supporting the generality of our findings.

Table 10: **Further experiments on GPT-4**

|                              | **Arithmetics** | **CSQA** | **HellaSwag** | **HH-RLHF** |
|------------------------------|-----------------|----------|---------------|-------------|
| Majority Voting              | **0.9967**      | 0.8721   | **0.9078**    | **0.5612**  |
| Decentralized MAD ($T = 1$)  | 0.9867          | **0.8788** | **0.9078**  | 0.5580      |
| Decentralized MAD ($T = 2$)  | 0.9867          | 0.8784   | 0.9044        | 0.5577      |
| Decentralized MAD ($T = 3$)  | 0.9833          | 0.8780   | 0.9044        | 0.5459      |

# H  Broader Impact, Limitations and Future Works

**Broader Impact.**  Our findings highlight an important perspective on Multi-Agent Debate, showing that much of its effectiveness can be achieved through simpler, more accessible methods like Majority Voting. This opens the door to building more efficient and scalable collaborative AI systems without sacrificing performance. Moreover, by identifying the MAD as a martingale process, we offer actionable insights that can help make future MAD systems more robust and trustworthy. We believe our work contributes to a new perspective on debate-based AI frameworks that are both principled and practical. Ultimately, this supports the broader goal of making AI systems more reliable, collaborative, and aligned with human reasoning.

**Limitations and Future Works.**  As discussed in Section 2, our study primarily focuses on the Simultaneous Talk protocol [3], where all agents generate and share their responses concurrently in each debate round. While this setting is widely adopted in prior work, it does not capture the full spectrum of possible communication strategies within multi-agent systems. Alternative protocols, such as One-by-One, where agents respond sequentially, or Simultaneous-Talk-with-Summarizer, where a summarizer agent oversees and summarizes the state of the debate, introduce different dynamics that may influence the rates of subversion and correction. Investigating these alternative protocols in depth remains an important direction for future work. Furthermore, our theoretical framework relies on the assumption of agent homogeneity and may not directly generalize to heterogeneous settings. A promising direction for future work is to extend the martingale analysis to account for heterogeneous agents with differing prior beliefs or reasoning capabilities.

