# OpenReview forum: "Debate or Vote: Which Yields Better Decisions in Multi-Agent Large Language Models?"
_NeurIPS.cc/2025/Conference — NeurIPS 2025 spotlight_

### Official Review · Reviewer_qiU5 · 2025-06-30

**Clarity:** 3
**Significance:** 2
**Originality:** 3
**Rating:** 4
**Confidence:** 4

**Summary:**

This paper investigates the core components of Multi-Agent Debate (MAD), arguing that its performance gains stem primarily from the aggregation of multiple agent outputs (i.e., Majority Voting) rather than the iterative debate process. While prior work has already raised empirical questions about the universal effectiveness of MAD compared to other ensembling methods [2] and has highlighted the power of voting protocols over extended debate rounds [3], this paper's main contribution is the introduction of a formal theoretical framework to explain why this is the case. The authors model the debate as a martingale process over agent beliefs, demonstrating that debate alone does not systematically improve expected correctness.

The modeling assumes that each agent follows a Dirichlet-Compound-Multinomial (DCM) process. The belief update rule is based on counting responses from neighboring agents. However, using simple count vectors for belief updates can make the complex interactions in LLM-based debates, and this modeling choice receives no further empirical validation. Only two observations inform the introduction of the assumption: the stochasticity of LLM outputs and the interaction between agents. The only supporting evidence is the stability of average accuracy over rounds, which is a minimal sign of the martingale behavior claimed.

The authors propose two MAD variants that bias belief toward correct responses. They observe locking correct answers, which breaks the martingale property and improves performance. Since this process needs access to ground truth, they propose heuristic methods using majority-voted responses. However, these methods often perform similarly to or worse than majority voting, limiting their practical value.

Overall, the paper presents a clean theoretical framework, but the modeling assumptions are strong and under-justified, and the proposed improvements offer limited advantage in practice.

[1] Tongxuan Liu, et al. "GroupDebate: Enhancing the Efficiency of Multi-Agent Debate Using Group Discussion." arXiv preprint arXiv:2409.14051 (2024)\
[2] Andries Smit, et al. "Should we be going MAD? A Look at Multi-Agent Debate Strategies for LLMs." Proceedings of the 41st International Conference on Machine Learning (2024)\
[3] Lars Benedikt Kaesberg, et al. "Voting or Consensus? Decision-Making in Multi-Agent Debate." arXiv preprint arXiv:2502.19130 (2025)

**Questions:**

**1**. The modeling assumes that belief updates can be represented by simple response counts, using the Dirichlet-Compound-Multinomial process. Can you provide empirical evidence or ablation results showing that this update rule reflects how real LLM responses evolve during debate?

**2**. The only support for the claimed martingale behavior is the stable mean accuracy over rounds. Did you check other signals such as $p_t$ directly or further support the theoretical claim?

**3**. The proposed methods that use majority-voted results often perform similarly to or worse than plain majority voting. How do you interpret this outcome, and under what conditions would your method provide clear advantages?

**Ethical Concerns:**

["NO or VERY MINOR ethics concerns only"]

**Final Justification:**

The authors have successfully addressed my primary technical concern by providing convincing empirical evidence for their theoretical framework; while practical limitations remain, I now recognize the significant value of their foundational contribution and have raised my score accordingly

**Limitations:**

yes

**Paper Formatting Concerns:**

There are no formatting concerns in this paper.

**Quality:**

3

**Strengths And Weaknesses:**

### **Strengths**

**Novel Theoretical Framework**: The primary strength of this paper is its novel theoretical framework. By modeling the multi-agent debate as a DCM process and martingale argument it provides a principled and rigorous explanation for why majority voting explains the success of MAD.

**Clarity and Organization**: The paper is exceptionally well-written and organized. The theoretical concepts, including the Dirichlet-Compound-Multinomial (DCM) model and the martingale proof, are presented with clarity, making the core arguments accessible.

### **Weaknesses**

**Limited Empirical Novelty**: The paper's central empirical conclusion, that majority voting is often as good as or better than complex debate while important, is not entirely novel. Recent studies have already shown that MAD is highly sensitive to hyperparameters and does not reliably outperform other prompting strategies [2]. Furthermore, the effectiveness of voting protocols and the potential negative effects of increasing debate rounds have been systematically demonstrated [3]. Thus, the paper's experiments seem as a confirmation of these existing findings rather than a new discovery.

**Strong and Underjustified Modeling Assumptions**: The theoretical framework, while elegant, rests on the strong assumption that the complex, semantic-rich interactions of LLM agents can be simplified to a Dirichlet-Compound-Multinomial (DCM) process updated by simple response counts. However, even standard prompt tuning stragies like reformulating question or refining response can adjust DCM $\alpha$ and marginal probability. The paper provides limited empirical validation that this model accurately captures the dynamics of real-world LLM debates, beyond showing that the average accuracy remains stable over rounds.

**Limited Practical Advantage of Proposed Methods**: The proposed interventions (e.g., MAD-Conformist), designed to break the martingale property, do not consistently or significantly outperform the simple majority voting baseline. This limits the practical impact of the theoretical insights, as the simpler, more efficient method remains the more compelling choice in most scenarios, a point also relevant in the context of improving MAD's efficiency [1].

---

> ### Author Rebuttal · Authors · 2025-07-29
>
> We thank the reviewer for the constructive feedback and for acknowledging the novelty of our theoretical framework, and also complimenting the clarity of our manuscript.
>
> Below, we address your key concerns.
>
> ---
>
> `A1` *Clarification on empirical finding*
>
>
> Thank you for this constructive feedback, and we appreciate the opportunity to better clarify the positioning of our work in relation to prior works.
>
> We agree that recent empirical studies—including Smit et al. [1] and Kaesberg et al. [2]—have made valuable contributions by revealing important limitations of multi-agent debate (MAD). These works raise fundamental questions about the robustness and efficacy of MAD, and we believe our paper offers a complementary and timely response to precisely address those concerns.
>
> Specifically, the goal of our empirical analysis is not to claim novelty in observing MAD's limitations, but rather to **motivate the need for a theoretical framework that can explain and predict when and why such limitations arise**. This is the central contribution and novelty of our work, as you also recognized. We develop a formal model of belief dynamics under debate and derive analytical results (e.g., Theorem 1 and 2) that shed light on how agent responses evolve and aggregate over time. This framework offers the **first principled explanation for this observation** -- something not accounted for in prior works. The value and novelty of our work are also recognized by peer reviewers:
>
> > "_I believe that the formulation introduced in the paper could have a moderate-to-high impact on the MAD community – a field which has suffered from imprecise and ad-hoc approaches in the past_." ---- Reviewer 6aic
>
>
> Lastly, we recognize that we should have cited and discussed these earlier works more directly in our motivation. We will revise the manuscript to better acknowledge their contributions, clarify how our work builds upon them, and make more explicit that our goal is to provide theoretical foundations for previously observed empirical patterns—thereby moving toward a deeper and more principled understanding of multi-agent systems.
>
> [1] Andries Smit, et al. "Should we be going MAD? A Look at Multi-Agent Debate Strategies for LLMs." Proceedings of the 41st International Conference on Machine Learning (2024)
>
> [2] Lars Benedikt Kaesberg, et al. "Voting or Consensus? Decision-Making in Multi-Agent Debate." arXiv preprint arXiv:2502.19130 (2025)
>
>
> ---
>
> `A2` *Clarification on modeling assumptions*
>
> Thank you for this thoughtful critique. We agree that modeling the rich and nuanced dynamics of LLM interactions is inherently challenging, and any abstraction—including ours—comes with limitations. Our goal is not to fully replicate the full semantic depth of LLM debates, but rather to **identify and formalize core statistical properties** that govern belief evolution and aggregation in multi-agent settings at a reasonable level of abstraction, and at the same time providing tractable and interpretable analysis.
>
> The Dirichlet-Compound-Multinomial (DCM) framework is an ideal choice, as we **justified in our manuscript (L150-L162)**:
>
> > "..._DCM closely mirrors how practical LLM systems produce different outputs due to uncertainty and sampling variability. In particular, the Dirichlet prior captures the agent’s internal belief over possible answers, while the Multinomial models the stochastic generation process (e.g., via temperature or nucleus sampling). This distribution is thus a natural choice because it encapsulates both internal uncertainty and output randomness, while also providing a principled Bayesian framework for belief updates across debate rounds—enabling analytical study of dynamics during the debate process...._"
>
>
> This proper abstraction enables us to derive formal insights about the role of belief dynamics, which **empirically align with key behavioral trends observed in real-world LLM-based MAD systems (more evidence discussed in `A3` below, where we directly verified $p_t$)**.
>
> We acknowledge that prompt tuning strategies----such as rephrasing or refinement----can influence how responses are generated and interpreted. And the sampling variation induced by prompting is precisely what DCM can capture mathematically. To account for such variation, **our framework is flexible enough to accommodate this by allowing a variable weight $w$ in the belief equation from Theorem 2**:
> $$
> p_t :=  \frac{\alpha_{i,t-1}^{(1)} + w^{(1)} \cdot c_{i,t}^{(1)}}{\sum_{j=1}^K ( \alpha_{i,t-1}^{(j)} + w^{(j)} \cdot c_{i,t}^{(j)})}.
> $$
> This weight can be adjusted via prompt design, fine-tuning, or human-guided signals, providing flexibility to model more realistic agent behaviors.
>
> We believe our heterogeneous-agent setup in Table 4 partially reflects this idea—for instance, a Programmer agent might be more influenced by the reasoning of a Mathematician than that of a Lawyer. Yet even in this more nuanced setting, performance trends remain largely stagnant, reinforcing the core idea of our theory.
>
> Overall, we view our theoretical framework as **a sufficiently general foundation that can incorporate richer dynamics and guide future improvements to debate protocols**.
>
> ---
>
> `A3` *Empirical evidence for DCM. Did you check signals such as $p_t$ directly?*
>
> Yes, we did! In fact, **Figure 4 presents the mean $p_t$ values across debate rounds** (with raw values provided in Appendix E). Importantly, $p_t$ reflects the single-agent belief in the correct answer at debate round $t$, and the mean $p_t$ is the average among agents ($y$-axis in the figure). This measure differs from the overall task accuracy, which the reviewer may have misinterpreted. This trajectory captures the agent's evolving belief and directly aligns with the DCM update model used in our theoretical analysis. As shown in Figure 4, **the mean $p_t$ values remain largely stationary over rounds, providing direct empirical support for the martingale behavior----consistent with the theoretical result in Theorem 2**.
>
> ---
>
>
> `A4` *Practical impact of theoretical insights*
>
> We appreciate the reviewer’s concern and would like to clarify on the practical impact of our theoretical analysis. One of the key insights of our analysis is that debate has the potential to substantially outperform simple voting when appropriately guided. This is clearly demonstrated through the MAD-Oracle. **These gains indicate that there is considerable headroom in the design space of MAD, and that our theory helps uncover this significant potential**.
>
> The purpose of our proposed interventions—MAD-Conformist and MAD-Follower—is not to fully bridge this gap, but to empirically validate concrete predictions derived from our theory. These methods are **intended to show promise**, demonstrating that even minimal, theory-driven modifications can improve upon vanilla MAD. **Our aim is not to claim state-of-the-art performance or argue against the utility of alternative approaches (including [1])**, but rather to establish a foundation for designing more effective strategies grounded in a deeper understanding of belief dynamics.
>
> We see these methods as **initial, illustrative steps**—proofs of concept that MAD performance can be steered by principled design. Far from limiting the practical value of our theory, these findings highlight its role in enabling targeted improvements. As we objectively and transparently note in the manuscript (L256–258):
>
> > "While they do not reach the oracle’s upper bound performance, they demonstrate that simple, theory-informed modifications can yield meaningful improvements—pointing to a promising direction for future work to close the gap."
>
> [1] Andries Smit, et al. "Should we be going MAD? A Look at Multi-Agent Debate Strategies for LLMs." Proceedings of the 41st International Conference on Machine Learning (2024)
>
>  ---
>
> `A5` *When will the interventions beat majority voting?*
>
> This is a very insightful question. The bottleneck in our current interventions is that they lack reliable estimation of which answer is correct, and thus cannot consistently introduce the directional bias needed to steer belief evolution productively. As a result, their gains are often modest. This stands in contrast to MAD-Oracle, which demonstrates the full potential of such a strategy when the correct signal is known/correctly estimated—achieving substantial improvements over majority voting.
>
> We expect that future work incorporating better estimators of correctness (e.g., via confidence scoring, calibrated self-reflection, or external signal integration) can realize significantly larger gains. In short, our framework suggests a clear direction for improving MAD: design agents that can more effectively estimate and amplify correctness early in the debate. We will incorporate such a discussion in our revised manuscript. Thanks again for your comment!

---

> > ### Comment · Reviewer_qiU5 · 2025-08-02
> >
> > Thank you for your response. I have carefully read your rebuttal and appreciate the clarifications provided. However, I must uphold my initial assessment, as the rebuttal does not adequately address my primary concerns. While I acknowledge the novelty of the theoretical framework, its empirical grounding and the practical implications of the proposed interventions are still weak.
> >
> > Below, I elaborate on why the rebuttal did not sufficiently address my key questions.
> >
> > **1. On Modeling Assumptions and Empirical Validation (Regarding Q1 & Q2):**
> >
> > My main critique was that the strong assumption of modeling LLM debate dynamics as a Dirichlet-Compound-Multinomial (DCM) process, updated by simple counts, lacks sufficient empirical validation. My question was a direct request for evidence beyond the stability of the mean belief.
> >
> > Your response to my question about the agent belief `p_i,t` is unsatisfying. I asked for a direct examination of **individual agent belief trajectories (`p_i,t`)** for each `i` to validate the martingale property, as the mean belief (`mean_i p_i,t`) can obscure significant individual-level fluctuations and is therefore not sufficient evidence. A stable average can easily result from diverging individual paths, which would contradict the underlying assumptions of your model.
> >
> > Your rebuttal simply pointed back to Figure 4, stating it shows the mean of `p_i,t` over `i` and claimed this is "direct empirical support." "This unfortunately sidesteps the statistical substance of my critique. Showing that the average is stable is not the same as showing that the process itself behaves like a martingale at the individual agent level.
> >
> > **Show us the direct `E[p_i,t]` for each i, not averaged over i from rerunning the experiments with randomness from each intermediate debate steps. If the belief propagation is really a martingale and is useful, it should also be consistent along the debate rounds.**
> >
> > **2. On the Practical Impact of Proposed Interventions (Regarding Q3 & Q4):**
> >
> > I appreciate your honesty in stating that the proposed methods (MAD-Conformist/Follower) are "proofs of concept" and do not reach the oracle's performance. However, the rebuttal does not change the fact that these interventions often perform similarly to or worse than the much simpler Majority Voting baseline, as shown in your own tables.
> >
> > My question was about the conditions under which your methods would provide a clear advantage. Your response points to future work with "better estimators of correctness." While this is a valid direction for future research, it confirms that in its current form, your proposed method's practical utility is limited and does not consistently outperform the baseline it aims to improve upon. The theoretical insight, while interesting, has not yet translated into a practically superior method.
> >
> > Therefore, I stand by my original rating. The work has potential, but it requires much stronger empirical support and a clearer demonstration of practical advantage to be a compelling contribution to the field.

---

> > > ### Author Response · Authors · 2025-08-02
> > >
> > > We thank the reviewer for the follow-up discussions!
> > >
> > > `A1` *Direct evaluation of $p_t$*
> > >
> > > To better illustrate the martingale property, we present the agent-wise $p_t$ values across debate rounds, for all datasets below.
> > >
> > > | Arithmetics  | Round 1 | Round 2 | Round 3 | Round 4 | Round 5 |
> > > |---------|---------|---------|---------|---------|---------|
> > > | Agent1  | 0.66    | 0.74    | 0.62    | 0.64    | 0.57    |
> > > | Agent2  | 0.66    | 0.73    | 0.67    | 0.65    | 0.67    |
> > > | Agent3  | 0.71    | 0.67    | 0.63    | 0.66    | 0.67    |
> > > | Agent4  | 0.70     | 0.60     | 0.66    | 0.65    | 0.65    |
> > > | Agent5  | 0.69    | 0.74    | 0.66    | 0.60     | 0.62    |
> > >
> > > | GSM8K       | Round 1 | Round 2 | Round 3 | Round 4 | Round 5 |
> > > |---------|---------|---------|---------|---------|---------|
> > > | Agent1  | 0.787   | 0.740    | 0.733   | 0.723   | 0.693   |
> > > | Agent2  | 0.793   | 0.727   | 0.727   | 0.727   | 0.720    |
> > > | Agent3  | 0.770    | 0.730    | 0.730    | 0.730    | 0.723   |
> > > | Agent4  | 0.780    | 0.753   | 0.713   | 0.717   | 0.690    |
> > > | Agent5  | 0.783   | 0.747   | 0.713   | 0.727   | 0.703   |
> > >
> > > | Pro. Medicine  | Round 1 | Round 2 | Round 3 | Round 4 | Round 5 |
> > > |---------|---------|---------|---------|---------|---------|
> > > | Agent1  | 0.765   | 0.765   | 0.732   | 0.721   | 0.710   |
> > > | Agent2  | 0.765   | 0.765   | 0.728   | 0.739   | 0.757   |
> > > | Agent3  | 0.794   | 0.743   | 0.702   | 0.706   | 0.724   |
> > > | Agent4  | 0.750   | 0.724   | 0.699   | 0.688   | 0.732   |
> > > | Agent5  | 0.768   | 0.754   | 0.739   | 0.724   | 0.724   |
> > >
> > > | Formal Logic | Round 1 | Round 2 | Round 3 | Round 4 | Round 5 |
> > > |---------|---------|---------|---------|---------|---------|
> > > | Agent1  | 0.373   | 0.389   | 0.405   | 0.405   | 0.444   |
> > > | Agent2  | 0.429   | 0.397   | 0.421   | 0.405   | 0.413   |
> > > | Agent3  | 0.381   | 0.373   | 0.405   | 0.468   | 0.397   |
> > > | Agent4  | 0.389   | 0.373   | 0.421   | 0.413   | 0.452   |
> > > | Agent5  | 0.341   | 0.389   | 0.373   | 0.397   | 0.437   |
> > >
> > > | HellaSwag  | Round 1 | Round 2 | Round 3 | Round 4 | Round 5 |
> > > |---------|---------|---------|---------|---------|---------|
> > > | Agent1  | 0.797   | 0.800   | 0.790   | 0.787   | 0.787   |
> > > | Agent2  | 0.803   | 0.810   | 0.793   | 0.793   | 0.773   |
> > > | Agent3  | 0.797   | 0.793   | 0.800   | 0.797   | 0.783   |
> > > | Agent4  | 0.800   | 0.807   | 0.790   | 0.800   | 0.777   |
> > > | Agent5  | 0.803   | 0.807   | 0.820   | 0.790   | 0.787   |
> > >
> > > | CSQA   | Round 1 | Round 2 | Round 3 | Round 4 | Round 5 |
> > > |---------|---------|---------|---------|---------|---------|
> > > | Agent1  | 0.780   | 0.773   | 0.733   | 0.757   | 0.740   |
> > > | Agent2  | 0.747   | 0.773   | 0.747   | 0.717   | 0.740   |
> > > | Agent3  | 0.730   | 0.730   | 0.750   | 0.747   | 0.730   |
> > > | Agent4  | 0.737   | 0.770   | 0.743   | 0.773   | 0.753   |
> > > | Agent5  | 0.737   | 0.740   | 0.740   | 0.753   | 0.737   |
> > >
> > > | HH-RLHF | Round 1 | Round 2 | Round 3 | Round 4 | Round 5 |
> > > |---------|---------|---------|---------|---------|---------|
> > > | Agent1  | 0.430   | 0.440   | 0.447   | 0.407   | 0.400   |
> > > | Agent2  | 0.477   | 0.443   | 0.430   | 0.410   | 0.403   |
> > > | Agent3  | 0.450   | 0.467   | 0.453   | 0.407   | 0.427   |
> > > | Agent4  | 0.430   | 0.430   | 0.410   | 0.407   | 0.397   |
> > > | Agent5  | 0.433   | 0.427   | 0.430   | 0.420   | 0.413   |
> > >
> > > ---
> > >
> > > `A2` *On the Practical Impact of Proposed Interventions*
> > >
> > >
> > > We would like to emphasize that the **proposed interventions (MAD-Conformist/Follower) are not the primary contribution of our work**. **A paper should be primarily judged based on the main contributions**, rather than the add-on explorations. Our primary contribution lies in our novel theoretical framework that characterizes the dynamics of multi-agent debate—an area that remains underexplored in the existing literature.
> > >
> > > The stronger evidence of the practical value of our framework comes from the MAD-Oracle experiment. This directly validates our theoretical prediction: when debate is properly biased toward the correct answer, it can substantially outperform majority voting, yielding up to **+17% average accuracy improvement**. The fact that such gains are achievable confirms that our framework does not just explain existing limitations, but also uncovers a large space of future improvements. **We see this as the clearest demonstration of practical impact**: while vanilla MAD often underperforms, the experiment shows that theory-guided debate design holds substantial untapped potential. **The impact of our work is therefore not in offering a single perfect intervention, but in charting a principled and clear direction where future advances can be made**.
> > >
> > > For these reasons, we believe the work makes a novel and impactful contribution to the community—an assessment echoed by multiple reviewers in their recognition of the framework’s originality and significance.

---

> ### Comment · Reviewer_qiU5 · 2025-08-04
>
> I thank the authors for providing the individual agent trajectories. However, their response appears to be a misunderstanding of my request.
>
> I intended to ask for an empirical validation of the martingale property. This requires estimating the conditional expected value, $\\mathbb{E}[p\_{i,t} | \\mathcal{F}\_{t-1}]$, for each agent `i` and time step `t`. Here, $\\mathcal{F}\_{t-1}$ denotes the filtration, representing all information available up to step `t-1`, including the agent's belief $p\_{i,t-1}$.
>
> **To estimate this expectation, one must**:
> 1.  Fix a state at an intermediate debate round (`t-1`).
> 2.  Rerun the subsequent step (`t`) multiple times to average over the stochasticity of the process.
>
> Presenting a single, random trajectory is insufficient to validate a claim about expected values, a point I have noted previously.
>
> To be perfectly clear, my request is to see direct evidence of whether the core martingale property, $\\mathbb{E}[p\_{i,t} | \\mathcal{F}\_{t-1}] \\approx p\_{i,t-1}$, holds empirically. **I strongly believe that, without this validation, the central theoretical claim of the paper lacks the necessary empirical support**.

---

> > ### Author Response · Authors · 2025-08-04
> >
> > Thank you for the follow-up question. We now fully understand your request and agree that it offers a more precise validation of the martingale property. As suggested, we compared the values of $p_{t-1}$ and $\mathbb{E}[p_t \mid \mathcal{F}\_{t-1}]$ using Llama 3.1 on Formal Logic and GSM8K. Here, $p_{t-1}$ denotes the belief of a randomly selected agent at round $t-1$, while $\mathbb{E}[p_t \mid \mathcal{F}_{t-1}]$ is computed by running 5 independent continuations of the debate and averaging the agent’s resulting $p_t$ values. Standard deviations are reported in parentheses.
> >
> > **Across the settings, we observe that $p_{t-1} \approx \mathbb{E}[p_t \mid \mathcal{F}\_{t-1}]$, directly confirming the martingale behavior in agent belief dynamics**:
> >
> > - Llama3.1 on MMLU-Formal Logic
> >
> > | round $t$ | $p_{t-1}$ | $\mathbb{E}[p_t \mid \mathcal{F}_{t-1}]$ |
> > |:---:|:---:|:---:|
> > | 1 | 0.388 | 0.380 (0.02) |
> > | 2 | 0.389 | 0.367 (0.02) |
> > | 3 | 0.392 | 0.395 (0.02) |
> >
> >
> > - Llama3.1 on GSM8K
> >
> > | round $t$ | $p_{t-1}$ | $\mathbb{E}[p_t \mid \mathcal{F}_{t-1}]$ |
> > |:---:|:---:|:---:|
> > | 1 | 0.686 | 0.686 (0.02) |
> > | 2 | 0.684 | 0.702 (0.01) |
> > | 3 | 0.694 | 0.695 (0.02) |
> >
> > We hope this targeted analysis addresses your remaining concern. Thank you again for the thoughtful follow-up and continued engagement!

---

> > > ### Comment · Reviewer_qiU5 · 2025-08-05
> > >
> > > Thank you for providing the additional experimental results in response to my repeated requests. The final analysis is convincing and effectively addresses my primary concern regarding the empirical validation of the martingale property.
> > >
> > > I now have a clearer understanding of your work's core contribution. I agree that the value of this paper lies in providing a novel theoretical framework to explain previously observed phenomena in MAD, which is a more rigorous contribution than previous approaches.
> > >
> > > While I still have reservations about the practical utility of the proposed interventions, I recognize the significance of the theoretical foundation that you have established. I will update my rating to reflect this.

---

> > > > ### Author Response · Authors · 2025-08-05
> > > >
> > > > Dear Reviewer qiU5,
> > > >
> > > > Thank you for actively engaging in the discussion and for your openness to re-evaluating our work. We are glad that your primary concern has been resolved. Your thoughtful questions and suggestions greatly helped strengthen the manuscript, and we truly appreciate your time and effort!
> > > >
> > > > Best regards,
> > > >
> > > > The Authors

---

### Official Review · Reviewer_6aic · 2025-06-30

**Clarity:** 4
**Significance:** 3
**Originality:** 3
**Rating:** 4
**Confidence:** 2

**Summary:**

The paper disentangles multi-agent debate (MAD) into two key components – Majority Voting and inter-agent Debate to assess the contribution arising from each. It demonstrates that the vast majority of the performance comes from Majority Votes, rather than from any agent debate. The paper goes on to introduce a mathematical framework and two theorems that help explain their empirical results, and highlight why in these homogeneous-agent Q&A tasks, inter-agent debate will not improve the expected correctness, i.e. Debate induces a martingale over agents’ belief trajectories, which in expectation remains unchanged across rounds.

**Questions:**

How do the empirical and theoretical findings presented in the paper align with paragraph “Pros of Multi-Agent Debate”?

For example, the conclusions of the theoretical analysis is that MAD induces “a martingale process, which preserves the expected success probability of each agent over time.” However, also stated on page 9: MAD “has the potential to enhance both factual accuracy and reasoning quality”. Please can you clarify how both these passages can hold?

**Ethical Concerns:**

["NO or VERY MINOR ethics concerns only"]

**Limitations:**

Yes

**Quality:**

3

**Strengths And Weaknesses:**

On Quality:

The paper presents strong motivation, and is well written. Clear and concise background is provided on the Majority Voting and Multi-Agent Debate, citing relevant works. It was a pleasure to read.

The paper does well to couple an empirical study (with 7 datasets and multiple strong LLM agents), with a theoretical formulation that explains the empirical findings. This is a particularly strong part of the paper, as to the best of my knowledge, many papers in this field do not formally define MAD with as much care as taken here. For example, careful attention is given to define MAD strategies as undirected graphs, where edges indicate which agents can observe one another.

The empirical evaluation is also conducted with care, and although the code is not released, it feels highly reproducible given the level of detail provided.

The Extended Experiments section preempted multiple of my questions – although a more thorough experiment into Open-Ended tasks would be welcomed (see below questions).
How would one consider extending the Bayesian update rule to account for the additional context an agent could provide to another, going beyond a simple count vector $c_{i,t}$ which would be required for open-ended tasks?

On Impact:

I believe that the formulation introduced in the paper could have a moderate-to-high impact on the MAD community – a field which has suffered from imprecise and ad-hoc approaches in the past.

However, I believe the paper can be more forthright with the assumptions made in the Theoretical Analysis section – which lead to the main conclusions of the paper.
For example, Line 172: “We assume that the correct answer has the largest belief $\alpha^{(1)}$”.
This feels like an overly strong assumption, and one that removes much of the
surprise of the findings.
Does this imply that the probability of the correct response is lower-bounded by Theorem 1 only in cases when the homogenous agent correctly assigns the largest $\alpha^{(1)}$ belief to that answer in round 0? If this is the case, are we surprised that majority voting provides the largest contribution to the aggregated response - given that the homogenous agent already “knows” the answer?

Surely the more interesting setting is the one where the underlying agent (or subset of agents) does not assign the largest belief to the correct response in the initial round – and how their belief can be updated over multiple rounds of debate? The theorem does not cover this more challenging case, and therefore one should be careful with the claims made, Line: 60: “we prove formally that majority vote does essentially all the work, which explains our empirical findings.”

Similarly, when we consider the agent’s Dirichlet parameter update, (Line 190) we are only considering the counts associated with the $K$ possible responses – this seems reductive and overly restrictive. For example, one would want an agent to be persuaded by the additional context and explanation another agent provides, biasing its belief away from its largest $\alpha$ value towards the correct response. Do you consider this setting?

For these reasons, it’s not clear to me that the theoretical results presented will hold when more complex and open-ended tasks are performed. How would a consensus even be defined in long form NPL tasks?

On Clarity:

Heterogeneous Agents: The manuscript would benefit from more context provided in this section. For example, what are “optimal persona sets” and how is this implemented in practice?
It would also be interesting to expand the theoretical analysis to this setting, but appreciate that might be future work.

---

> ### Author Rebuttal · Authors · 2025-07-29
>
> We thank the reviewer for the constructive and thoughtful feedback! We are deeply encouraged by and grateful for your words:
>
> > "_I believe that the formulation introduced in the paper could have a moderate-to-high impact on the MAD community – a field which has suffered from imprecise and ad-hoc approaches in the past_."
>
> which is exactly what prompted us to write this paper :)
>
> Below, we address your key concerns.
>
>
> ---
>
> `A1` *Assumption that the correct answer has the largest initial belief $\alpha^{(1)}$*
>
> You are raising an excellent question here. We would like to emphasize that **this assumption reflects only a weak prior preference**, not a strong or certain belief. For example, suppose a single agent assigns a belief of 0.01 to the correct answer and 0.009 to the next-best option. While the agent’s accuracy is only 1%, Theorem 1 shows that with enough agents, majority voting can boost correctness to nearly 100%, which is the core benefit of majority voting. Importantly, this works even if the correct option is only slightly favored, no matter how small that margin is.
>
> Moreover, **we provide an alternative version of the theorem in Appendix D, which does not assume that the correct option has the largest initial belief**. Theorem 1.A establishes a lower bound on the probability that the correct answer is selected by majority vote, without assuming any belief constraints. This highlights that the benefit of voting is not solely due to prior confidence, but also emerges from the statistical structure of ensembling.
>
>
>
> Lastly, we want to emphasize that **this assumption is not required for the general formulation of our belief dynamics or for understanding the behavior of multi-agent debate more broadly**. Theorem 2 does not rely on this assumption. The martingale property holds regardless of whether the correct option initially has the highest belief or not. It tells us that in expectation, beliefs do not drift toward the correct answer over time unless external structure or directional bias is introduced. This explains why naive multi-agent debate may not consistently outperform majority voting—debate alone, without guidance, cannot correct a misaligned initial prior.
>
> Indeed, the most important and challenging setting is where the agents do not initially favor the correct answer. **Our theoretical framework helps us understand what structural mechanisms are needed to shift beliefs in such settings**. It also highlights the limitations of relying solely on peer interactions and motivates our design of guided interventions.
>
> ---
>
> `A2` *Extending the Bayesian update rule beyond count vectors and defining "consensus" for open-ended tasks*
>
> This is an insightful and important question! We agree that multi-agent debate becomes more nuanced in open-ended or long-form NLP tasks. That said, we believe the core intuition of our belief update framework remains applicable, with a few key adaptations. Below, we address both (i) extending the update rule and (ii) defining consensus in open-ended tasks:
>
> **(i) Extending the update rule**
>
> Although open-ended responses are not easily countable in a strict categorical sense, they can still be modeled in distributional or similarity-based forms. Specifically, the core mechanism of our Bayesian update rule,
> $$
> p_t :=  \frac{\alpha_{i,t-1}^{(1)} + c_{i,t}^{(1)}}{\sum_{j=1}^K ( \alpha_{i,t-1}^{(j)} + c_{i,t}^{(j)})},
> $$
> does not inherently require hard categorical responses. In open-ended settings, the count vector $c_{i,t}$ can be interpreted more generally—for example, as
>
> - a soft histogram over clustered response types,
> - an embedding-based semantic agreement metric,
> - or a weighted similarity score between textual outputs.
>
> Thus, **the Bayesian belief update process can be extended by letting $c_{i,t}$ reflect aggregated semantic signals** rather than strict response counts.
>
> **(ii) Defining consensus in open-ended tasks**
>
> Similar to the extended update rule for open-ended tasks, **consensus is better defined semantically, not symbolically**. For example, if agents independently generate similar explanations or rationales, we may consider this as semantic consensus, even if the surface forms differ. This can be operationalized via thresholds on embedding similarity, Levenshtein distance, or overlap in reasoning chains etc.
>
> Our current experiments use discrete agreement on multiple-choice question tasks, but we believe the above adaptations can extend naturally to open-ended contexts. We view this direction as an exciting next step for generalizing the MAD framework beyond classification.
>
> ---
>
> `A3` *Limiting counts to $K$ possible responses is restrictive ... one would want an agent to be persuaded by the additional context from other agents, biasing its belief towards the correct response. Do you consider this setting?*
>
> Absolutely! We agree that persuasion via explanation is a desirable property of debate. However, we respectfully clarify that the limitation to $K$ discrete responses is not inherently at odds with such persuasive dynamics.
>
> The use of $K$ responses in our framework is simply a modeling abstraction to make the belief update process tractable and interpretable. This $K$-way formulation can be readily generalized to larger or even infinite response spaces (e.g., open-ended generation tasks discussed in Q1), without invalidating the core martingale result. What matters is how agents map peer responses—whether discrete or continuous—into belief adjustments.
>
> ---
>
> `A4` *What are “optimal persona sets” and how is this implemented in practice?*
>
> As noted in Lines 272–276, the "optimal persona sets" refer to a collection of agent roles selected via the "agent selection algorithm" introduced in [1]. This method identifies a combination of personas that are empirically effective when collaborating in multi-agent tasks.
>
> In practice, we implement this by assigning each agent a system prompt that encodes a specific role or persona—e.g., Mathematician, Programmer, Lawyer—using the same prompt templates provided in [1]. We agree that our manuscript would benefit from more explicit implementation details, and we will include a full description of the relevant details in the revised version.
>
> [1] Z. Liu, et al. "Dynamic llm-agent network: An llm-agent collaboration framework with agent team optimization" (COLM 2024).
>
> ---
>
> `A5` *How do the empirical and theoretical findings presented in the paper align with paragraph “Pros of Multi-Agent Debate”?*
>
> Thank you for this observation. First, we would like to clarify that the paragraph titled "Pros of Multi-Agent Debate" appears in the Related Works section and is intended as a summary of claims made in prior literature—not the central claim of our own work.
>
> In contrast, our paper critically examines those assumptions and challenges the notion that multi-agent debate in its current form consistently improves factual accuracy or reasoning quality. Our theoretical and empirical results show that, without carefully designed mechanisms or external interventions, MAD can stagnate or even degrade performance over time due to its martingale dynamics.
>
> We appreciate the opportunity to clarify this and will revise the manuscript to clearly distinguish the claims of prior work from our contributions.

---

> > ### Comment · Reviewer_6aic · 2025-08-07
> >
> > Thank you for your detailed response.
> >
> > `A1` In the example you gave:
> > > suppose a single agent assigns a belief of 0.01 to the correct answer and 0.009 to the next-best option
> >
> > Does this assume that no other answer is given a belief higher than 0.01? i.e. 0.01 is the maximum value in the belief vector?
> > If so, the comments in my original review hold. If not, please clarify.
> >
> > I appreciate the additional theorem which does not make this assumption -- which I believe is overly restrictive in practice.
> >
> >  `A2-3` Super interesting extensions - thanks for clarifying.
> >
> > `A4-5` Thanks for the additional detail. I believe some of this should make it into the manuscript.
> >
> > Ultimately, I believe my score reflects my review.

---

> > > ### Author Response · Authors · 2025-08-08
> > >
> > > Thank you for taking the time to read our response! We are glad that our response has addressed your concerns, and that you find some of the extensions interesting. We answer your follow-up question below.
> > >
> > > `A1` *0.01 is the maximum value in the belief vector?*
> > >
> > > Yes your understanding is correct. Theorem 1 shows that, regardless of how small this value is, majority voting can magnify its probability up to 100% given a sufficient number of agents.
> > >
> > > We do agree with your view and thus have provided the alternative theorem in Appendix, **which does not rely on such an assumption**.
> > >
> > > We hope this clarifies all your questions, and we thank you again for your support and engagement in the discussion. We will make sure to include those additional details in the manuscript.
> > >
> > > Best,
> > >
> > > The Authors.

---

### Official Review · Reviewer_hHcS · 2025-07-01

**Clarity:** 3
**Significance:** 4
**Originality:** 3
**Rating:** 5
**Confidence:** 3

**Summary:**

This paper explores the effectiveness of Multi-Agent Debate in improving the performance of large language models. It contrasts MAD with simple Majority Voting through extensive experiments across seven NLP benchmarks. The key findings are that Majority Voting accounts for most of the performance gains typically attributed to MAD, and that debate alone does not systematically improve expected correctness. The paper also presents a theoretical framework modeling debate as a stochastic process and proves that it induces a martingale over agents’ belief trajectories. Based on these insights, the authors propose interventions to enhance debate effectiveness by biasing belief updates toward correct signals.

**Questions:**

1. Why is the Dirichlet-Compound-Multinomial distribution chosen for modeling LLM response generation? Are there related papers or statistical data supporting this choice?

2. What is the base model used for Figure 3?

3. Can an average result column be added to Table 1 and Table 2?

**Ethical Concerns:**

["NO or VERY MINOR ethics concerns only"]

**Final Justification:**

The authors have provided responses to the questions I raised earlier regarding the Dirichlet-Compound-Multinomial distribution and the verification of model stability in their rebuttal. After rechecking, I confirm that my overall score of this paper remains consistent with the assessment conclusions presented in the initial review.

**Limitations:**

Yes

**Paper Formatting Concerns:**

None.

**Quality:**

3

**Strengths And Weaknesses:**

Strengths:

1. This paper puts forward an interesting and significant argument that Majority Voting is responsible for most of the performance gains usually associated with MAD, and it substantiates this argument through both experimental and theoretical analysis.

2. The paper compares Majority Voting and various MAD techniques across over seven NLP datasets, and the experimental results largely align with the theoretical proof.

3. The conclusions drawn in this paper hold certain implications for enhancing the design of MAD architectures.

Weaknesses:

1. The paper mainly conducts comparative experiments on the Qwen2.5-7B-Instruct and Llama3.1-8B-Instruct base models. The experimental results on Llama3.1-8B-Instruct show a certain degree of instability, and experiments on larger-scale bases are only performed on some benchmarks, making it impossible to fully understand the model's patterns. Additionally, the authors do not test on other closed-source LLMs.

2. The conclusions of the paper mainly target closed-ended question answering tasks, and the applicability of Majority Voting techniques to open-ended text generation tasks is limited. Whether there is a Majority Voting architecture suitable for text generation tasks remains a topic of discussion.

---

> ### Author Rebuttal · Authors · 2025-07-29
>
> We thank the reviewer for the constructive feedback and for **acknowledging the value of our experiments and theoretical analyses**.
>
> Below, we address the key concerns.
>
> ---
>
> `A1` *Instability of LLaMA3.1-8B-Instruct results*
>
> Thank you for the question! We would like to emphasize that this behavior is entirely consistent with our theoretical framework, which models MAD as a martingale process—meaning that the expected correctness of agents does not improve with additional debate rounds unless directional signals are injected.
>
> As shown in Table 1 in the manuscript, LLaMA occasionally shows gains from MAD, but those gains are modest and non-monotonic, aligning with our martingale-based explanation (Theorem 2). This demonstrates that instability is not a flaw of the experiment, but a key empirical support for our theoretical claim: MAD, in its vanilla form, does not consistently lead to correctness improvements. We will clarify this interpretation more explicitly in the revised manuscript.
>
> ---
>
> `A2` *Further experiments on larger-scale models*
>
> Thank you for this suggestion. In response, we have conducted additional experiments using the larger-scale Qwen2.5-32B-Instruct model across four representative benchmarks. As shown in the table below, the effect of Decentralized MAD varies. For Arithmetics, GSM8K, and HellaSwag, we observe slight performance degradation or oscillation across rounds, while for Professional Medicine (MMLU), MAD improves over majority voting. These results further support the generality of our martingale-based analysis: even with highly capable agents.
>
>
> |  | Arithmetics | GSM8K | Pro. Medicine | HellaSwag |
> |:---:|:---:|:---:|:---:|:---:|
> | Majority Voting | 1.000 | 0.9433 | 0.9007 | 0.8767 |
> | Decentralized MAD ($T = 2$) | 0.9733 | 0.9367 | 0.9154 | 0.8767 |
> | Decentralized MAD ($T = 3$) | 0.9667 | 0.9200 | 0.9154 | 0.8733 |
> | Decentralized MAD ($T = 5$) | 0.9700 | 0.9300 | 0.9191 | 0.8667 |
>
> ---
>
> `A3` *Further experiments on closed-source models*
>
> In response to your request for experiments on closed-source models, we conducted additional evaluations using three GPT-4 agents across four benchmarks. The overall trends remain consistent with those observed in our open-source model experiments, supporting the generality of our findings.
>
> |   | Arithmetics | CSQA | HellaSwag | HH-RLHF |
> |:---:|:---:|:---:|:---:|:---:|
> | Majority Voting | 0.9967 | 0.8721 |0.9078 | 0.5612 |
> | Decentralized MAD ($T = 1$) | 0.9867 | 0.8788 | 0.9078 | 0.5580 |
> | Decentralized MAD ($T = 2$) | 0.9867 | 0.8784 | 0.9044 | 0.5577 |
> | Decentralized MAD ($T = 3$) | 0.9833 | 0.8780 | 0.9044 | 0.5459 |
>
> ---
>
>
> `A4` *Application to open-ended tasks*
>
> Great point! We agree that multi-agent debate becomes more nuanced in open-ended or long-form NLP tasks. That said, we believe the core intuition of our belief update framework remains applicable, with a few key adaptations. In particular, we discuss below: (i) extending the update rule and (ii) defining consensus in open-ended tasks:
>
> **(i) Extending the update rule**
>
> Although open-ended responses are not easily countable in a strict categorical sense, they can still be modeled in distributional or similarity-based forms. Specifically, the core mechanism of our Bayesian update rule,
> $$
> p_t :=  \frac{\alpha_{i,t-1}^{(1)} + c_{i,t}^{(1)}}{\sum_{j=1}^K ( \alpha_{i,t-1}^{(j)} + c_{i,t}^{(j)})},
> $$
> does not inherently require hard categorical responses. In open-ended settings, the count vector $c_{i,t}$ can be interpreted more general—for example, as
>
> - a soft histogram over clustered response types,
> - an embedding-based semantic agreement metric,
> - or a weighted similarity score between textual outputs.
>
> Thus, **the Bayesian belief update process can be extended by letting $c_{i,t}$ reflect aggregated semantic signals** rather than strict response counts.
>
> **(ii) Defining consensus in open-ended tasks**
>
> Similar to the extended update rule for open-ended tasks, **consensus is better defined semantically, not symbolically**. For example, if agents independently generate similar explanations or rationales, we may consider this as semantic consensus, even if the surface forms differ. This can be operationalized via thresholds on embedding similarity, Levenshtein distance, or overlap in reasoning chains etc.
>
> Our current experiments use discrete agreement on multiple-choice question tasks, but we believe the above adaptations can extend naturally to open-ended contexts. We view this direction as an exciting next step for generalizing the MAD framework beyond classification.
>
> ---
>
> `A5` *Rationale for choosing the Dirichlet-Compound-Multinomial model for LLM response generation*
>
> This is an insightful question. We adopt the Dirichlet-Compound-Multinomial (DCM) formulation as the backbone of our belief-update model, grounded in both theoretical rigor and empirical observations:
>
> - **Principled Bayesian foundation**: The Dirichlet distribution is the conjugate prior of the Multinomial distribution, making it a straightforward and well-established choice for modeling the nature of LLM text generation dynamics. In our setup, an agent’s belief over $K$ possible responses is updated as it receives signals from other agents, and the DCM provides a closed-form way to model this update: an agent starts with a Dirichlet prior $\alpha$, observes peer responses, and updates beliefs based on the posterior. This mirrors classical Bayesian updating and lends a principled probabilistic structure to belief evolution in a multi-agent setting.
>
> - **Empirical alignment with LLM dynamics**: As shown in Figure 4 of our manuscript, the evolution of $p_t$ over rounds of decentralized debate aligns well with the expectation-neutral behavior predicted by DCM. That is, belief trajectories exhibit symmetric drift under majority influence, without systemic bias toward any particular answer choice. This symmetry and neutrality are core predictions of DCM and match what we observe across multiple rounds of real LLM-agent interactions.
>
> ---
>
> `A6` *What is the base model used for Figure 3?*
>
> The base model for Figure 3 is Qwen2.5-7B-Instruct. We will make sure to include this in our revised version. Thank you for pointing this out.
>
> ---
>
>
> `A7` *Can an average result column be added to Tables 1 and 2?*
>
> Of course! We will add the average columns in our revised version. We appreciate your suggestion.

---

> > ### Comment · Reviewer_hHcS · 2025-08-05
> >
> > Thanks for the authors' responses. Most of my concerns have been addressed. I will keep my score which I think is in line with my review.

---

> > > ### Author Response · Authors · 2025-08-05
> > >
> > > Dear Reviewer hHcS,
> > >
> > > Thank you for the response and for your positive feedback. We are glad that most of your concerns have been addressed! We sincerely appreciate your time and effort in engaging in the review process.
> > >
> > > Best,
> > >
> > > The Authors

---

### Official Review · Reviewer_HZhH · 2025-07-03

**Clarity:** 3
**Significance:** 3
**Originality:** 3
**Rating:** 4
**Confidence:** 4

**Summary:**

In this paper, the authors breakdown MAD into two components, majority voting and inter-agent debate. Through extensive experiments across 7 NLP benchmarks, they show that majority voting is responsible for most, if not all ,of the improvement seen in MAD techniques vs single agent numbers.

To explain this, they develop a theoretical framework to model the debates as a stochastic, peer-influenced process. They prove that an unguided debate process induces a martingale over the agents' belief trajectories, i.e. expected correctness does not improve withe debate alone. Empirical results in the paper, support this as well.

Based on this, the paper proposes MAD-Conformist and MAD-Follower which use majority votes to guide the belief update process to bias the belief update process towards correctness. Both methods show a strong improvement over standard MAD techniques across a variety of tasks.

**Questions:**

In a majority of the results, it seems that correctness scores decrease with more debate rounds. Do the authors have an insight into why that is the case? Is it due to the degradation of initial strong majorities or a few agents changing their response causing weaker majorities to fade? Is it an overall breakdown of agreement, i.e. as the debate progresses, the models disagree with each other more rather than agree?

**Ethical Concerns:**

["NO or VERY MINOR ethics concerns only"]

**Final Justification:**

I appreciate the explanation on how the framework allows for diversity in the debate and the lexical diversity analysis of the different personas.

**Limitations:**

Yes

**Quality:**

3

**Strengths And Weaknesses:**

### Strengths
 - This paper does a great job of breaking down why MAD does and doesn't work as expected. It provides strong empirical proof for majority voting being the largest contributor to any MAD improvements and develops a sound mathematical explanation for why that is the case. Modeling the agents as DCMs and the debate process as a martingale provides helps understand how MAD works.
 - The paper's claims are substantiated by a comprehensive set of experiments. It multiple models of different sizes and model families, diverse benchmarks (from reasoning to QA), and various MAD configurations (decentralized, sparse, etc.), which strongly supports the generalizability of the findings.
 - The paper includes simple but impactful improvements to MAD, incorporating majority voting to bias the agents' belief update process towards correctness.
 - This paper shows that sometimes simpler is better, and serves a reminder of the crucial role a good baseline plays in contextualizing improvements.


### Weaknesses
 - The theoretical framework and most empirical results are based on homogenous agents. While this is useful for simplicity, it somewhat defeats the purpose of a debate. Since all agents share the same beliefs, there is less variety in the responses generated by them. [Estornell et. al.](https://proceedings.neurips.cc/paper_files/paper/2024/file/32e07a110c6c6acf1afbf2bf82b614ad-Paper-Conference.pdf) show that a lack of diversity in responses or model capabilities is detrimental to the debate process.
- The lack of diversity is somewhat addressed through the heterogenous agents experiment in section 6 but the models are all still similarly capable. It would also be helpful to include a t-SNE visualization of the response embeddings to confirm that the diverse personas result in truly diverse responses.
 - The experimental setup is based on a relatively simple MAD design (simultaneous-talk, simple debate). The paper doesn't evaluate against more sophisticated methods involving structured debates, where the results might not generalize.
 - Limited novelty for MAD-Conformist and MAD-Follower: [Smit et. al.](https://arxiv.org/pdf/2311.17371) propose an agreement modulation mechanism to precisely control what proportion of responses agree with each other and show that that high inter-agent agreement leads to improved correctness.  The paper would be more impactful if it contextualized its proposed methods against similar related work.

---

> ### Author Rebuttal · Authors · 2025-07-29
>
> We thank the reviewer for the constructive feedback and for complimenting the strong and extensive empirical results.
>
> Below, we address the key concerns.
>
> ---
>
> `A1` *Clarification on agent setting*
>
> We understand the reviewer’s concern and fully agree that diversity is valuable in multi-agent debate settings. We would like to clarify the key considerations of out setup:
>
> - Homogeneous agents allow us to **isolate the core mechanisms**, rigorously dissecting the contribution of inter-agent *debate* vs. simple *majority voting*, without introducing confounding factors such as varied model capabilities or role-specific behaviors. This clean setup enables both a rigorous theoretical analysis and a principled understanding of where performance gains originate.
> - The homogeneous agent setting is the most commonly studied setup in prior MAD work (e.g., [1,2,3,4,5]), **making our theoretical framework broadly relevant and well-grounded in the established research landscape**. **For a theoretical foundation paper, it's important to first fully understand the essential ground**, which can later inform and extend to more advanced/specialized cases.
>
> That said, we want to emphasize that **our framework does not preclude diversity**:
>
> 1. **Stochastic response diversity does arise in our setting**: As Estornell et al. rightly emphasize, response diversity is important for productive debate. Crucially, _such diversity can still emerge in homogeneous agent setups_. Even when agents share the same base prompt and belief prior, stochastic sampling (e.g., temperature tuning or top-p sampling) induces variability in responses—allowing agents to explore different reasoning paths and generate disagreement.
> 2. **Our theory does allow capturing different beliefs**: As defined in Definition 1 of our paper, each agent samples its belief $\boldsymbol{\theta}\_{i,t}$ from a Dirichlet distribution. Even if all agents share the same prior concentration parameter $\boldsymbol{\alpha}$, the sampled beliefs $\boldsymbol{\theta}\_{i,0}$ differ across agents due to the inherent randomness in the Dirichlet process. This means that agents can hold distinct initial beliefs and produce diverse responses, enabling non-trivial interactions during debate.
> 3. Lastly, **our theoretical model and empirical results remain applicable and informative, even as richer forms of agent heterogeneity are introduced**. As you recognized, we *do* study heterogeneous agents in Section 6 (Table 4), where we assign different personas (e.g., "Mathematician", "Lawyer") following established practices. Even in these diverse settings, we observe that simple Majority Voting remains highly competitive, further supporting our claim that ensembling is a primary driver of MAD performance. For these reasons, we believe our work lays a strong foundation for future research in this direction.
>
>
> [1] C. Chan et al., "ChatEval: Towards Better LLM-based Evaluators through Multi-Agent Debate" (ICLR 2024)
>
> [2] Y. Du et al., "Improving Factuality and Reasoning in Language Models through Multiagent Debate" (ICML 2024)
>
> [3] Y. Li et al., "Improving Multi-Agent Debate with Sparse Communication Topology" (EMNLP-findings 2024)
>
> [4] Q. Wang et al., "Rethinking the Bounds of LLM Reasoning: Are Multi-Agent Discussions the Key?" (ACL 2024)
>
> [5] G. Zhang et al., "Cut the Crap: An Economical Communication Pipeline for LLM-based Multi-Agent Systems" (ICLR 2025)
>
>
> ---
>
> `A2` *Do diverse personas result in diverse responses?*
>
> We appreciate the reviewer’s interesting suggestion to plot the t-SNE projections. We looked further into this, but since persona is not an explicit attribute that is easily captured in output embeddings, we found that t-SNE projections do not show meaningful differences between single- and multi-persona setups.
>
> As an alternative, we retrieved the frequency table of the mean Levenshtein distance, which is computed as:
>
> $$
> D = \frac{1}{N^2}\sum_{i=1}^N\sum_{j=1}^N \text{Levenshtein}(y_i, y_j),
> $$
>
> where $y_i$ is the response from agent $i$ (see table below). 30 samples from GSM-8K were evaluated on the Qwen2.5 agents. The table reveals a clear shift under the multi-persona configuration, **indicating greater lexical diversity in the generated responses**.
>
> | **bins** | **Single-persona** | **Multi-persona** |
> |:---:|:---:|:---:|
> | $350 \leq D <400$ | 13 | 1 |
> | $400\leq D < 450$ | 11 | 16 |
> | $450 \leq D < 500$ | 6 | 10 |
> | $500 \leq D < 550$ | 0 | 2 |
> | $550 \leq D < 600$ | 0 | 1 |
>
> ---
>
> `A3` *Structured debate methods*
>
>
> We appreciate the reviewer’s point. Structured debate formats (role-based, judge-mediated systems, etc.) are indeed a powerful line of work. Our focus, however, is to understand the fundamental dynamics of inter-agent communication, which is why we adopt a minimal, decentralized protocol to isolate core mechanisms without added structure.
>
> That said, **our theoretical framework is not restricted to simplistic settings**. **The DCM-based model applies broadly to any debate mechanism** where agents update beliefs based on peer responses. Notably, the belief update rule in Theorem 2 can be better generalized as:
> $$
> p_t :=  \frac{\alpha_{i,t-1}^{(1)} + w^{(1)} \cdot c_{i,t}^{(1)}}{\sum_{j=1}^K ( \alpha_{i,t-1}^{(j)} + w^{(j)} \cdot c_{i,t}^{(j)})},
> $$
> which includes a flexible weight term $w$ that can capture structured signals—such as those from prompt design, fine-tuning, or human-guided adjudication. Structured systems like judge-based debate can be seen as increasing $w^{(1)}$ for more credible or correct responses, introducing a directional bias that helps overcome the stagnation highlighted in our model.
>
> We also want to highlight that **other reviewers have explicitly recognized the impact and foundational value of our framework**:
>
> > “_I believe that the formulation introduced in the paper could have a **moderate-to-high impact** on the MAD community – a field which has suffered from imprecise and ad-hoc approaches in the past_.” — *Reviewer 6aic*
>
>
> In summary, **our framework offers a general foundation for understanding belief dynamics, and can serve as a diagnostic tool even in structured settings**. By revealing when and why debate fails to improve accuracy, it provides diagnostic insight that can guide the design of more effective coordination strategies. We see integration with structured protocols as a valuable direction for future work.
>
> ---
>
> `A4` *Novelty of MAD-Conformist and MAD‑Follower—needs better context like agreement modulation (Smit et al.).*
>
> We thank the reviewer’s suggestion to better contextualize our interventions within related work. We acknowledge that Smit et al.’s work on agreement modulation is closely related and highlights an important connection to our proposed interventions. Our interventions—MAD-Conformist and MAD-Follower—are theoretically motivated by our martingale analysis (Theorem 2), aiming to address the core stagnation issue in MAD dynamics through lightweight, generalizable mechanisms.
>
> To further clarify positioning, we will explicitly compare our designs with related strategies such as Agreement Modulation (Smit et al.), All-Agents Draft (AAD) and confidence-weighted voting (Kaesberg et al.) in the revised paper. We believe this framing will highlight the novelty and complementary nature of our work, both theoretically and practically.
>
> Thank you again for this constructive suggestion, which we believe strengthens the paper.
>
> ---
>
> `A5` *Why does accuracy often decrease with more debate rounds?*
>
> According to our theoretical framework (Theorem 2), belief updates under a DCM process are expectation-neutral; i.e., in the absence of external directional signals, agents are equally likely to drift toward or away from the correct answer over successive debate rounds. However, **when generalizing the belief update rule (see `A3`), it is possible that the dynamics become skewed**: if the update weight $w$ is larger for incorrect responses—e.g., due to persuasive but flawed reasoning, the expected correctness can decline over time.
>
> To capture this phenomenon, we analyze the subversion and correction rates of Qwen2.5 agents across debate rounds (see table below). Here, subversion rate denotes the proportion of responses that were correct in the previous round but became incorrect in the next, while correction rate tracks the opposite. Overall, the subversion rate seems to overwhelm correction rates, suggesting that real-world LLM agents may be slightly more frequently persuaded away from the correct answer—an empirical indicator that the weights $w$ assigned to incorrect responses are higher in practice.
>
> | **Dataset** | **Round** | **Correction Rate (%)** | **Subversion Rate (%)** |
> |:---:|:---:|:---:|:---:|
> | Arithmetics | 2 | 0.40 | 2.00 |
> | | 3 | 0.00 | 1.80 |
> | | 5 | 0.20 | 3.00 |
> | GSM8K | 2 | 0.33 | 1.33 |
> | | 3 | 0.40 | 1.73 |
> | | 5 | 0.40 | 2.13 |
> | Pro. Medicine | 2 | 0.96 | 0.96 |
> | | 3 | 0.96 | 1.10 |
> | | 5 | 0.96 | 1.03 |
> | Formal Logic | 2 | 1.43 | 1.90 |
> | | 3 | 1.90 | 2.38 |
> | | 5 | 1.75 | 2.06 |
> | HellaSwag | 2 | 0.33 | 0.40 |
> | | 3 | 0.33 | 0.73 |
> | | 5 | 0.40 | 1.20 |
> | CSQA | 2 | 0.73 | 1.00 |
> | | 3 | 0.53 | 1.40 |
> | | 5 | 0.60 | 1.13 |
> | HH-RLHF | 2 | 1.33 | 2.13 |
> | | 3 | 1.40 | 2.53 |
> | | 5 | 1.27 | 2.53 |

---

> ### Author Response · Authors · 2025-08-04
>
> Dear Reviewer HZhH,
>
> Thank you for your follow-up and for recognizing the additional analysis we provided. We sincerely appreciate your openness to reassessing the submission. Your revised evaluation is encouraging, and we’re grateful for your time and effort in reviewing our responses!
>
> Best regards,
>
> The Authors

---

### Note · Authors · 2025-08-11

Dear Reviewers, AC, SAC and PC,

We extend our sincere gratitude for the time and effort you have dedicated to reviewing our manuscript.

In this paper, **we establish a new theoretical foundation that characterizes the dynamics of multi-agent debate.**
We are grateful for the reviewers’ recognition of the novelty, theoretical rigor, and empirical soundness of our contributions.
For convenience, we summarize below the key strengths and concerns noted by the reviewers, along with our responses:

---
**Key Strengths noted by the reviewers and our responses:**
- **S1:** A **novel and sound theoretical framework** for explaining why MAD often fails, and how majority voting accounts for most of the performance gains. (All Reviewers)
- **S2:** **Impactful formulation for the MAD community**, including meaningful implications for future MAD architecture design.  (Reviewers HZhH, hHcS, 6aic)
- **S3:** A **well-described and extensive experimental setup**, exploring diverse agent configurations. (All Reviewers)
- **S4:** **Exceptional writing quality** (Reviewers 6aic, qiU5)
---
**Key concerns and how we addressed them**:
- **C1:** *Clarification on the modeling assumptions and the generalizability of its Bayesian update rules.* We explained our rationale for choosing the DCM model and provided a **flexible interpretation of the update rule** that extends to heterogeneous agent settings and open-ended tasks. We also added **direct empirical verification** of the debate dynamics. (All Reviewers)
- **C2:** *Implications of theory.*  We clarified that our intervention (MAD-oracle) directly shows the practical impact and **uncovers a principled path for future advances in MAD design**. While additional interventions (MAD-follower/conformist) are not our primary contribution, they are illustrative initial attempts in that direction. (Reviewer HZhH, qiU5)
- **C3:** *Need for additional experiments.* We included further experiments on **larger models** (e.g., Qwen2.5-32B-Instruct) and on **closed-source models** (e.g., GPT-4). (Reviewer hHcS)
---
We would also like to note that **we have thoroughly addressed all four reviewers' concerns to their satisfaction**. Following the rebuttal, our manuscript has received broad support for acceptance. We will incorporate their comments and the additional discussion into the final manuscript, further strengthening its quality.

Thank you very much for your time and effort. We hope this summary is helpful.

Best regards,

Authors.

---

### Decision · Program_Chairs · 2025-09-17

**Decision:**

Accept (spotlight)

**Comment:**

* **Summary**:
  * Ablation study on Multi-Agent Debate components over 7 NLP benchmark.
  * Show that majority voting is the main driver of improvement over single-agent.
  * Develop theoretical framework to model debates as stochastic processes.
  * Prove that process is a martingale over beliefs: correctness does not improve with debate alone. Back this proof with empirical experiments.
  * Propose two new MAD strategies using majority votes to guide the belief update process towards correctness: both show strong improvement over variety of tasks.
* **Strengths:**
  * Great break down of why MAD does and doesn’t work. (HZhH, hHcS)
  * Strong empirical proof (HZhH, hHcS)
  * Sound mathematical explanation (HZhH, hHcS)
  * Novel theoretical formalization it introduces is much-needed in the MAD community (6aic, qiU5)
  * Theory backed by comprehensive set of experiments, varying model sizes, families, benchmarks tasks, MAD configurations, supporting generalizability (HZhH, hHcS, 6aic)
  * Provides simple but impactful improvements on MAD (HZhH, hHcS)
  * Pleasure to read (6aic), Exceptionally well-written and organized (6aic)
  * Empirical conducted with care and feels highly reproducible thanks to level of detail provided (6aic)
  * Extended experiments preempted multiple questions of the reviewer (6aic)
  *
* **Weaknesses:**
  * Relies on homogenous agents, which might be over-simplistic. (HZhH, 6aic) → Convincingly covered in rebuttal.
  * Simple MAD design: would be curious to evaluate on more sophisticated MAD designs involving structured debate (HZhH) → convincingly covered in rebuttal.
  * Instability of result in experiment with LLaMA (hHcS) → Convincingly explained as a feature, not a bug, in the rebuttal
  * Could better compare its new MAD strategies against similar work (HZhH) → Added in rebuttal.
  * Would appreciate more LLMs and more variability at high-scale LLMs, and closed-source LLMs (hHcS) → Both added in rebuttal.
  * Question about modeling choices / assumptions (HzhH, hHcS, qiU5, wxsp) → Discussed thoroughly in rebuttal.
  * Limited novelty of discovery that majority voting is as good as or better than debate (wxsp) → Acknowledged in rebuttal, as novelty is in theoretical framework that offers first explanation of this observation.
  * New modified MAD does not clearly outperform majority voting (wxsp) → acknowledged in rebuttal, shows “proof of concept”
* **Summary of rebuttal:**
  * Discussion of Simplicity of homogenous agents (HzhH) used to isolate core mechanism, as a foundational theoretical paper.
  * Theory still holds true with diversity factored in.
  * Will add comparison to similar work suggested by (HzhH)
  * Thorough rebuttal with generalization of formulas and extra evaluation.
  * Raised (HzhH) score.
  * Explained LLaMa experiment’s instability as backed by their theory (hHCs)
  * Added high-scale experiments (hHCs)
  * Added closed-source experiments (hHCs)
  * Convincing discussion of the chosen distribution model (hHCs, 6aic)
  * Discussion of modeling assumptions in the theorem, with alternative provided without that assumptions and clarification thereof in the manuscript (6aic, wxsp)
  * Extension of update rule (6aic)
  * Surprisingly, reviewer (6aic) chose to keep his 4 score in spite of what I (as AC) see as a thorough answer by the authors.
  * Added references (wxsp)
  * Multiple rounds of discussion with (wxsp), adding new data from experiments, with clarifications as to how experiments validate some precise points of the theoretical framework. → Authors add new experiments showing it, convinces (wxsp) who updates their score.
* **Missing:**
  * Would be interesting to discuss generalization beyond close-ended questions, on open-ended text generation (hHcS) → Elaborated in rebuttal.
* **Most important reason for accept/reject**: On a red-hot topic, a great combination of empirical analysis of a problem, introduction of a much-needed novel formalization, paired with thorough empirical validation, and an exemplary interaction with the reviewers adding abundant new experiments and clarifications that convinced all of them. Thought-provoking, timely, and beautiful\!
* **Why stands out:** Our field desperately needs proper, thorough theoretical analysis such as this one, which directly connect to practice by backing it with empirical evidence such as this one.